# Measurement of the vertical atmospheric density profile from the X-ray Earth occultation of the Crab Nebula with *Insight*-HXMT

Daochun Yu [1,3], Haitao Li [1,2,3], Baoquan Li [1,2,3], Mingyu Ge[4], Youli Tuo[4], Xiaobo Li[4], Wangchen Xue[3,4], Yaning Liu[1,2], Aoying Wang[5], Yajun Zhu[1,3,6], and Bingxian Luo[1,3,7]

[1]National Space Science Center, Chinese Academy of Sciences, Beijing 100190, China

[2]Key Laboratory of Electronics and Information Technology for Space Systerms, Chinese Academy of Sciences, Beijing 100190, China

[3]University of Chinese Academy of Sciences, Chinese Academy of Sciences, Beijing 100049, China

[4]Key Laboratory of Particle Astrophysics, Institute of High Energy Physics, Chinese Academy of Sciences, Beijing 100049, China

[5]School of Physical Science and Technology, Lanzhou University, Lanzhou 730000, China

[6]State Key Laboratory of Space Weather, Beijing 100190, China

[7]Key Laboratory of Science and Technology on Environmental Space Situation Awareness, Chinese Academy of Sciences, Beijing 100190, China

**Correspondence:** Haitao Li (lihaitao@nssc.ac.cn) Baoquan Li (lbq@nssc.ac.cn)

**Abstract.** The X-ray Earth occultation sounding (XEOS) is an emerging method for measuring the neutral density in the lower thermosphere. In this paper, the X-ray Earth occultation (XEO) of the Crab Nebula is investigated by using the Hard X-ray Modulation Telescope (*Insight*-HXMT). The pointing observation data on the 30th September, 2018 recorded by the Low Energy X-ray telescope (LE) of *Insight*-HXMT are selected and analyzed. The extinction lightcurves and spectra during the X-ray Earth occultation process are extracted. A forward model for the XEO lightcurve is established and the theoretical observational signal for lightcurve is predicted. The atmospheric density model is built with a scale factor to the commonly used MSIS density profile within a certain altitude range. A Bayesian data analysis method is developed for the XEO lightcurve modeling and the atmospheric density retrieval. The posterior probability distribution of the model parameters is derived through the Markov Chain Monte Carlo (MCMC) algorithm with the NRLMSISE-00 model and the NRLMSIS 2.0 model as basis functions and the best-fit density profiles are retrieved respectively. It is found that in the altitude range of 105–200 km, the retrieved density profile is 88.8% of the density of NRLMSISE-00 and 109.7% of the density of NRLMSIS 2.0 by fitting the lightcurve in the energy range of 1.0–2.5 keV based on XEOS method. In the altitude range of 95–125 km, the retrieved density profile is 81.0% of the density of NRLMSISE-00 and 92.3% of the density of NRLMSIS 2.0 by fitting the lightcurve in the energy range of 2.5–6.0 keV based on XEOS method. In the altitude range of 85–110 km, the retrieved density profile is 87.7% of the density of NRLMSISE-00 and 101.4% of the density of NRLMSIS 2.0 by fitting the lightcurve in the energy range of 6.0–10.0 keV based on XEOS method. Goodness-of-fit testing is carried out for the validation of the results. The measurements of density profiles are compared to the NRLMSISE-00/NRLMSIS 2.0 model simulations and the previous retrieval results with NASA's Rossi X-ray Timing Explorer (RXTE) satellite. For further confirmation, we also compare the measured density profile to the ones by a standard spectrum retrieval method with an iterative inversion technique. Finally, we find that the retrieved density profile from *Insight*-HXMT based on the NRLMSISE-00/NRLMSIS 2.0 models is qualitatively

consistent with the previous retrieved results from RXTE. The results of lightcurve fitting and standard energy spectrum fitting are in good agreement. This research provides a method for the evaluation of the density profiles from MISIS model predictions. This study demonstrates that the XEOS from the X-ray astronomical satellite *Insight*-HXMT can provide an approach for the study of the upper atmosphere. The *Insight*-HXMT satellite can join the family of the XEOS. The *Insight*-HXMT satellite with other X-ray astronomical satellites in orbit can form a space observation network for XEOS in the future.

## 1  Introduction

The middle and upper atmosphere is affected by both solar activity and geomagnetic disturbance, so it is of great significance to study the density of the middle and upper atmosphere for further understanding solar-terrestrial relationships (Rhoden et al., 2000; Prölss, 2011). The middle and upper atmosphere is the passage area of reentry vehicle. In addition, the middle and upper atmosphere density is extremely important as an input of aerodynamic force and heating design of the reentry vehicle (Riley and Dejarnette, 1992; Davis and White, 2008), although it is very thin. In addition, maneuver planning, precise orbit determination and satellite lifetime predictions are limited by the accuracy of neutral density in the thermosphere (Doornbos and Klinkrad, 2006; Kalafatoglu Eyiguler et al., 2019).

However, the measured data of the density in the middle and upper atmosphere are scarce, especially in the upper mesosphere and lower thermosphere (60–200 km), due to the limitation of detection methods (Russell et al., 1999; Baron et al., 2020; Zeitler et al., 2021). With the increasing demand for the density of the Earth's middle and upper atmosphere, various semi-empirical atmosphere models have been developed, such as Jacchia Reference Atmosphere (Jacchia, 1970, 1977), Drag Temperature Model (DTM) (Berger et al., 1998; Bruinsma et al., 2003) and Mass Spectrometer Incoherent Scatter Radar Extended model (MSIS) (Hedin, 1987; Picone et al., 2002) from Naval Research Laboratory. Generally, there are still errors of around 30% RMS and peak errors of 100% or more in those semi-empirical models due to the complex changes in the middle and upper atmosphere (Doornbos et al., 2008). Some new methods have been developed to detect the density of the Earth's middle and upper atmosphere.

Originally, in-situ measurements are used to obtain the density of the middle and upper atmosphere, which is a way to obtain the atmosphere density directly by payloads from sounding rockets. In-situ measurements of atmospheric density near orbit can be achieved by using satellites. As an in-situ measurement method, falling sphere measurements can also provide the vertical atmospheric density profiles (Bartman et al., 1956; Faire and Champion, 1965; Faucher et al., 1967; Haycock et al., 1968). The nitric oxide density profile between 60 and 96 km was deduced from measurements by a sounding rocket (Pearce, 1969). The local-noon mean ozone distribution was obtained by analyzing the data from 21 sounding rockets carrying the Arcas optical ozonesondes up to 52 km (Krueger, 1973). Direct measurements of atmospheric density at high altitudes, especially above 100 km, using sounding rocket methods are difficult because of the short duration of rocket flights and their high cost (Watanabe, 1958). China completed the atmospheric density detection and precise orbit determination (APOD) mission with four CubeSats, which was designed to estimate atmospheric density below 520 km using in-situ sounding and precision orbit products and to demonstrate the link between geomagnetic storms and density enhancements (Tang et al., 2020). In order to

obtain the spatiotemporal variation characteristics of atmospheric density on a global scale, the remote sensing by satellites is gradually developed. As a scientific instrument of Thermosphere Ionosphere Mesosphere Energetics and Dynamics (TIMED) that is the initial mission under NASA's Solar Terrestrial Probes Program, Sounding of the Atmosphere Using Broadband Emission Radiometry (SABER) can obtain vertical profiles of several atmospheric constituents, such as $O_3$, $H_2O$ and $CO_2$, as well as neutral atmospheric density in the altitude range of $\sim$10–140 km (Russell et al., 1999; Meier et al., 2015; Rezac et al., 2015).

In addition to the direct measurements of atmospheric density by sounding rockets, satellites and falling sphere measurements, retrieval of atmospheric density can also be carried out by an indirect method that usually refers to the method of occultation sounding. Stellar occultation has a long history as an atmospheric diagnostic method. The technique of retrieval of atmospheric density by occultation is gradually developed. There are some previous studies on the retrieval of atmospheric density of specific species by stellar occultation in the ultraviolet band. Hays and Roble (1973) obtained the nighttime vertical distribution of ozone number density in an altitude range of 60–100 km at low latitudes by analyzing the results of approximately 12 stellar occultations in the ultraviolet band near 2500 Å. Aikin et al. (1993) measured the molecular oxygen densities in the altitude range of 140–220 km based on the solar occultation data obtained from the ultraviolet spectrometer/polarimeter (UVSP) on the Solar Maximum Mission (SMM) spacecraft. The density profiles of ozone and nitrogen dioxide were inverted and evaluated by an optimal estimation algorithm using solar occultation data from SCanning Imaging Absorption spectroMeter for Atmospheric CHartographY (SCIAMACHY) in the UV-Vis wavelength range (Meyer et al., 2005). Lumpe et al. (2007) used an optimal estimation algorithm to obtain the $O_2$ density profiles between 110 and 240 km by analyzing solar occultation data at three nominal wavelengths (144, 161 and 171 nm). In addition to relevant studies in ultraviolet band, occultation in infrared and radio band has also been extensively studied. The water vapour number density profiles in the altitude range of 15–45 km was retrieved by the solar occultation data with SCIAMACHY in the wavelength region around 940 nm (Noël et al., 2010). The radio occultation technique can be often used to retrieve the electron density in the ionospheric by the Abel integral equations (Hajj and Romans, 1998; Lei et al., 2007; Chou et al., 2017). There are also occultation measurements that retrieve atmospheric density for specific species, such as SOFIE/AIM (McHugh et al., 2008; Rong et al., 2016), GOMOS/Envisat (Renard et al., 2008; Kyrölä et al., 2010), SAGE series (Degenstein et al., 2018; McCormick et al., 2020) and POAM series (Rusch et al., 2001; Lumpe et al., 2002).

The study of atmosphere by X-ray occultation is a new interdisciplinary study. The atmospheric extent of Titan was measured by the transit of the Crab Nebula in the X-ray band on 2003 January 5 observed by the Chandra X-Ray Observatory (Mori et al., 2004). Rahmati et al. (2020) obtained the neutral density of the Mars atmosphere in the altitude range of 50–100 km by the Martian atmosphere occultation of 10 keV X-ray of Scorpius X–1 used the SEP instrument on the MAVEN satellite. The X-ray Earth occultation technique is also a unique method to retrieve the neutral atmospheric density in the upper mesosphere and lower thermosphere. Based on the X-ray occultation of the Crab Nebula with ARGOS/USA and the X-ray occultation of Cygnus X–2 with RXTE/PCA, Determan et al. (2007) obtained the Earth's atmospheric density in the altitude ranges of 100–120 km and 70–90 km. Very recently, the Earth's average atmospheric density at low latitudes in the altitude range of 70–200 km was measured by analyzing 219 X-ray Earth occultations data of the Crab Nebula with Suzaku and Hitomi (Katsuda et al.,

2021). However, the retrieved atmospheric densities are significantly lower than the model density in some altitude ranges, and the difference between the measured values and the model values may result from the long-term accumulation of greenhouse gases, imperfect climatological estimates of solar and geomagnetic effects, temperature profile differences or gravity waves (Determan et al., 2007; Katsuda et al., 2021). Therefore, it is very important to cross-check the density structure of Earth's atmosphere by observations from other X-ray satellites like *Insight*-HXMT, to further verify the difference between retrieved results and model density.

The X-ray Earth occultation sounding (XEOS) has many advantages as an atmospheric diagnostic method. The X-ray photons are absorbed directly by the K-shell and L-shell electrons of atoms, including atoms within molecules, in the extinction process. Therefore, the ionized states, electronic states and chemical bonds within the molecules of atmospheric components have no effect on the absorption of X-rays. XEOS can retrieve the neutral atmospheric density in the upper mesosphere and lower thermosphere. In addition, the global distribution of neutral atmosphere in the upper mesosphere and lower thermosphere can be obtained by analyzing a large number of X-ray Earth occultation data from X-ray satellites, and the temporal evolution characteristics of neutral atmosphere density in this altitude range can also be studied. In this work, we use the *Insight*-HXMT observational data to study the X-ray Earth occultation of the Crab Nebula in order to demonstrate the capability of *Insight*-HXMT as an atmospheric diagnostic instrument. In addition, the retrieved results of *Insight*-HXMT and RXTE are used to cross-check the density structure of Earth's atmosphere, so as to confirm the existence of differences between retrieved results and model values. Here, the theoretical model of lightcurve is established by simulating the observations of the Low Energy X-ray telescope (LE) (Chen et al., 2020) to the Earth occultation of the Crab Nebula. A Bayesian data analysis framework is developed for the XEO lightcurve modeling and the atmospheric density retrieval. We use Markov Chain Monte Carlo (MCMC) algorithm to calculate the posterior probability distribution of model parameters, which is a method of inverting model parameters using Bayesian inference (Sharma, 2017). Finally, the Earth's atmospheric density in the altitude range of 85–200 km is retrieved.

The paper is structured as follows. Sect. 2 describes the observations and data reduction. Sect. 3 shows lightcurve modeling and density profile retrieval. The conclusions and discussions are given in Sect. 4.

## 2   Observations and data reduction

*Insight*-HXMT is the first X-ray astronomy satellite in China (Zhang et al., 2018; Li et al., 2018; Zhang et al., 2020). It is designed for the three main scientific objectives including Galactic Plane scanning, X-ray binaries observation, Gamma-Ray Bursts and Gravitational Wave Electromagnetic counterparts monitoring and studying (Zhang et al., 2020). The *Insight*-HXMT mainly carries four scientific payloads, including the High Energy X-ray telescope (HE), the Medium Energy X-ray telescope (ME), the Low Energy X-ray telescope (LE) and the Space Environment Monitor (SEM) (Zhang et al., 2018, 2020). The quality of calibration directly determines the achievement of the three main scientific objectives. The Crab Nebula is one of the brightest X-ray sources in the sky, with its stable evolution and brightness. Therefore, the Crab Nebula is an excellent calibration source for many X-ray satellites. The Crab Nebula as a standard candle has been widely used for in-flight calibration

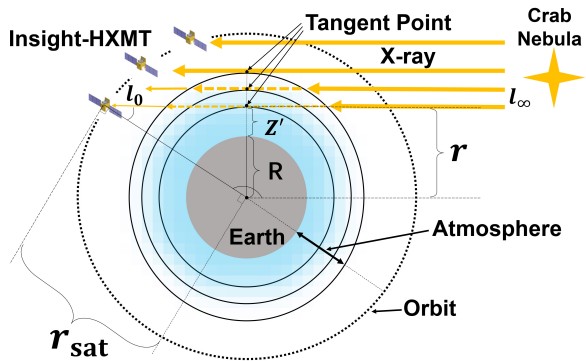

**Figure 1.** (Color online) The observation geometry of the X-ray Earth occultation of the Crab Nebula with *Insight*-HXMT. Tangent point altitude ($Z'$) is the shortest distance from the Earth's surface to the line of sight. The line of sight is defined as the line from the satellite position $l_0$ to the X-ray source position $l_\infty$. $r_{\text{sat}}$ is the distance from the satellite to the center of the Earth. $r$ is the distance from the tangent point to the center of the Earth. $R$ is the radius of the Earth. The dotted line shows the satellite orbit. The extent of the atmosphere is marked by a double sided arrow. The location of the tangent point is marked. The *Insight*-HXMT and the Crab Nebula are also marked. For clarity, only three layers of the atmosphere are marked, as shown by the black solid lines. The thick, orange solid lines with arrows show the X-rays before absorption and scattering. The orange dashed lines with arrows show the absorption and scattering of X-rays by the atmosphere. The thin, orange solid lines with arrows show X-rays passing through the atmosphere.

of space missions in X-ray astronomy (Kirsch et al., 2005; Meyer et al., 2010). In this work, the Crab Nebula is chosen as our observation source. The pointed observation mode is selected for the study of the X-ray Earth occultation.

## 2.1 Observation geometry

As Crab Nebula sets behind or rises from the limb of Earth as seen by *Insight*-HXMT, the X-ray flux from the Crab Nebula detected by *Insight*-HXMT varies due to the absorption of X-ray photons by Earth's atmosphere. The observation geometry of the X-ray Earth occultation of the Crab Nebula with *Insight*-HXMT is shown in Figure 1. In the process of the X-ray Earth occultation, the atmosphere reduces the flux of the X-rays detected by *Insight*-HXMT. As the line of sight moves closer to the Earth's surface (setting), more X-ray photons are absorbed by the atmosphere, and vice versa.

## 2.2 Data reduction

The observation data of the Low Energy X-ray telescope (LE) are used in this study. We select the photons observed by detectors for LE because the extinction effect at lower energy band during the occultation is obvious due to the larger cross section of the Earth's upper atmospheric compositions (Figure 2). LE consists of three detector boxes each containing 32 pieces of CCD236 (Chen et al., 2020) that is the second-generation Swept Charge Device (SCD) designed for X-ray spectroscopy (Holland and Pool, 2008; Zhao et al., 2019). The collimators divide each detector box of LE into four kinds of fields of view (FOV). For each detector box, 20 CCD236 have small FOVs of $1.6°\times6°$, 6 CCD236 have wide FOVs of $4°\times6°$, 2 CCD236

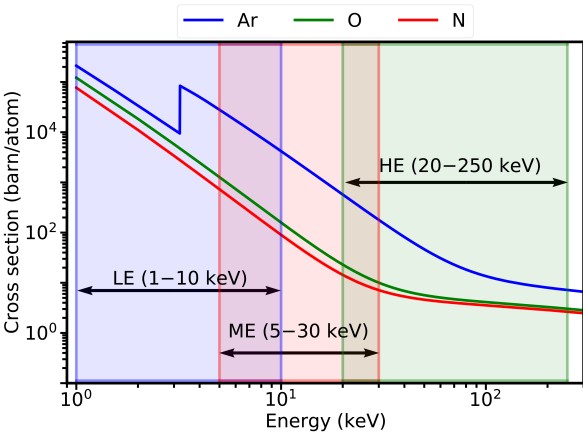

**Figure 2.** (Color online) X-ray cross sections for Ar, O and N. The energy ranges of LE, ME and HE are indicated by shaded areas in light blue, light red and light green to better distinguish the cross sections of each energy range. LE, ME, HE and their respective energy ranges are also marked by bidirectional arrows. The calibrated energy range of LE is shown. A barn is a unit of area where 1 barn=1 cm$^{-24}$.

**Table 1.** Summary of X-ray Earth occultation of the Crab Nebula analyzed in this study.

| Obs ID | Target | Right ascension (Ra) | Declination (Dec) | Start time (UTC) | Stop time (UTC) | Occultation type |
|--------|--------|----------------------|-------------------|------------------|-----------------|------------------|
| P0101299007 | Crab | 83.6330° | 22.0145° | 2018-09-30 15:38:36 | 2018-09-30 15:42:17 | Egress |

are blind FOVs and 4 CCD236 have large FOVs of about 50°–60°×2°–6° (Zhao et al., 2019; Chen et al., 2020; Zhang et al., 2020). The detector response matrix is generated through calibration database hxmt CALDB (v2.05)[1]. Only observations from the small FOV detectors excluding the detector ID of 29 and 87 that are damaged are used for analysis in order to get accurate background in this paper.

The information of the observation data selected in the data reduction is listed in Table 1, including the observation ID, the start and end time of the observation, the target source, and the right ascension and declination of the source in the coordinate system J2000. In the data reduction process, by using HXMTsoft(v2.04)[2], we extract the lightcurves and spectra of the Crab Nebula during Earth occultation recorded with LE. For the LE instrument, the photon counts are recorded by CCD236 detectors. Since we mainly study the occultation process of the Crab Nebula by the Earth atmosphere, the good time interval is screened by the following criteria: *ELV* (the elevation of the pointing direction above the horizon) less than 10°.

### 2.3 Description of spectra and lightcurves

The comparison of the X-ray energy spectra during the occultation process is shown in Figure 3. Five X-ray spectra in different altitude ranges are shown for clarity. These five energy spectra in blue, red, orange, magenta and green in Figure 3 are derived

---

[1]http://hxmtweb.ihep.ac.cn/caldb/628.jhtml

[2]http://hxmtweb.ihep.ac.cn/software.jhtml

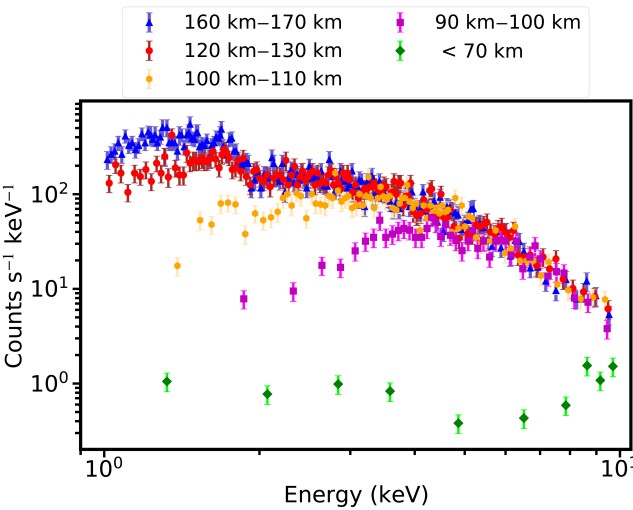

**Figure 3.** (Color online) The comparison of the X-ray energy spectra during the occultation process. These spectra cover an energy range of 1–10 keV (0.1240 –1.2398 nm). The unattenuated X-ray energy spectrum is shown in blue. The red, orange and purple data points are the attenuated X-ray energy spectra with the decreasing tangent point, respectively. The energy spectrum in green is the fully attenuated X-ray spectrum.

from the results of subsamples at altitudes of 160–170 km, 120–130 km, 100–110 km, 90–100 km and below 70 km, starting from an unattenuated energy spectrum to the partially attenuated energy spectrum, and ending with a fully attenuated energy spectrum. It is shown that the flux of the X-ray energy spectrum attenuates with reducing tangent point altitude during the occultation. Moreover, the absorption of X-ray photons by the atmosphere decreases with increasing energy.

In the process of X-ray Earth occultation of the Crab Nebula, the Crab Nebula and Earth's atmospheric disks are tangent at four contact times $t_I$–$t_{IV}$, illustrated in Figure 4. The total duration is $t_T = t_{IV} - t_I$, the full duration is $t_F = t_{III} - t_{II}$, the ingress duration is $t_o = t_{II} - t_I$, and the egress duration is $t'_o = t_{IV} - t_{III}$. The occultation depth $\delta$ is the X-ray flux attenuation due to the extinction.

The egress duration is about 14 seconds during the occultation process analyzed in this study. We divide this duration into 35 bins in order to have good time resolutions and high signal-to-noise ratio (SNR). The lightcurves of three different energy bands during the occultation process are shown in Figure 5. The abscissa time is converted to tangent point altitude. In this study, the maximum height difference between two adjacent tangent points is 836 meters, and the mean height difference between two adjacent tangent points is about 673 meters. For clarity, the data points in Figure 5 are displayed by taking one point every ten points from the initial data points. The dashed lines of blue, red and green represent the modelled lightcurves. The green and blue shadow colored regions correspond to the extinction process for the occultation in the energy bands of 1.319 keV–1.725 keV and 7.006 keV–7.412 keV, respectively. For clarity, the height range for occultation between 3.350 keV and 3.756 keV are not marked. The lightcurve in the energy range of 1.319 keV–1.725 keV starts to attenuate at 150 km and it

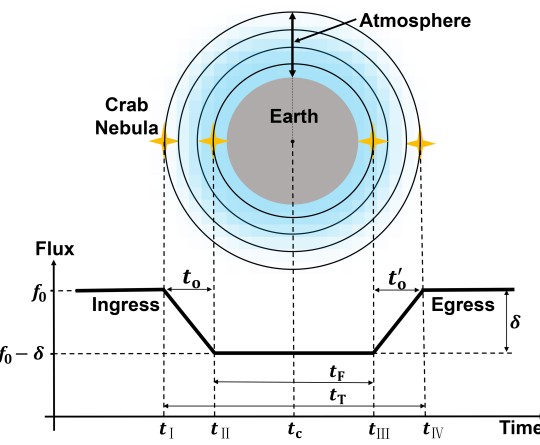

**Figure 4.** (Color online) Illustration of an X-ray Earth occultation of the Crab Nebula, showing the lightcurve discussed in Sect. 2.2, the four contact points $t_I$, $t_{II}$, $t_{III}$, $t_{IV}$ and the quantities $t_o$, $t_F$, $t_T$, $t_c$ and $\delta$ defined in Sect2.3. As a Low earth orbit (LEO) satellite, *Insight*-HXMT can observe the Crab Nebula twice in one orbit (egress and ingress). The length of $t_F$ is half the orbital period minus the duration of two occultations, and the orbital period of *Insight*-HXMT is about 96 minutes.

is completely attenuated at 102 km. The lightcurve in the energy range 7.006 keV–7.412 keV starts to attenuate at 100 km and it is completely attenuated at 85 km.

## 3 Lightcurve modeling and density profile retrieval

In this section, we will describe the details of the lightcurve modeling for the X-ray Earth occultation of the Crab Nebula. In this study, we model the Earth occultation as a measurement method for atmospheric density.

X-rays can be absorbed by the photoelectric effect. The ionized states, electronic states and chemical bonds within the molecules of atmospheric components have no effect on the absorption of X-rays in the extinction process. X-ray photons are absorbed directly by the K-shell and L-shell electrons of atoms, including atoms within molecules. Therefore, the X-ray Earth occultation can work as an atmospheric diagnostics method. In this case, the source celestial coordinates and the satellite positions are known, whereupon the atmospheric density profile can be treated as the unknown. It is impossible to distinguish atoms from molecules (in the calculation process, although X-ray photons interact directly with the K- and L- shells electrons of atoms (including atoms within molecules), the $O_2$ (or $N_2$) counts as one absorbing "particles" in the calculation, so the total neutral atmospheric density profile can be retrieved by X-ray occultations (Katsuda et al., 2021).

The schematic of lightcurve modeling of the X-ray Earth occultation is shown in Figure 6. The attenuation process of the X-ray energy spectrum can be described by Beer-Lambert law during the occultation. The attenuation energy spectrum is convolved with the detector response matrix to obtain the forward model. Given the data and the forward model, Bayesian inference is used to estimate the model parameters. Given the prior distribution and the likelihood function, the posterior probability distribution of the model parameters is calculated by MCMC. The best fit model is obtained from the posterior

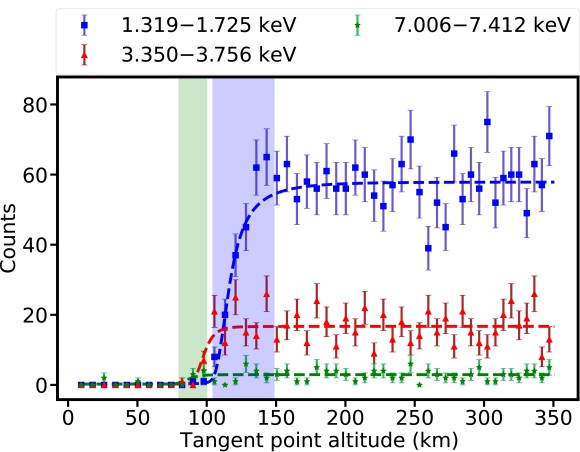

**Figure 5.** (Color online) The lightcurves during the occultation process. The points with the error bars are observation data. The dashed lines of blue, red and green represent the modelled lightcurves. The green shadow colored region corresponds to the height range for the occultation in the energy bands of 7.006 keV–7.412 keV. The height range for occultation in the energy range of 7.006 keV–7.412 keV is about 85–100 km. The blue colored region corresponds to the height range for the occultation in the energy bands of 1.319 keV–1.725 keV. The height range for occultation in the energy range of 1.319 keV–1.725 keV is about 102–150 km. For clarity, the height range for occultation between 3.350 keV and 3.756 keV are not marked. The height range for occultation in the energy range of 3.350 keV–3.756 keV is about 90–110 km. The reason for the relatively large variability of each lightcurve is the absorption of X-ray photons by atoms of atmospheric constituents.

probability distribution of the model parameters. The results will be compared with other measurements and models. The details of the data analysis will be described in the following subsection.

### 3.1 Forward model

The attenuation process of X-rays in the atmosphere can be described by Beer-Lambert law

$$I = I_0 e^{-\tau}, \tag{1}$$

where $I_0$ is the unattenuated source spectrum, which is a function of energy, $e^{-\tau}$ is the transmittance, $\tau$ is the optical depth, which has the form

$$\tau = \sum_s \gamma \int_{l_0}^{\infty} n_s \sigma_s dl, \tag{2}$$

where $s$ labels the gas components in the Earth's atmosphere, $\gamma$ is the total correction factor, $n_s$ is the number density of each component of the atmosphere along the line of sight. Based on the spherical symmetry assumption of the Earth atmosphere, the number density is converted to column density by Abel integral. $\sigma_s$ is the X-ray cross section (photoelectric absorption and scattering cross section) of each component in the atmosphere. X-ray photons are absorbed or scattered by atoms, and in the

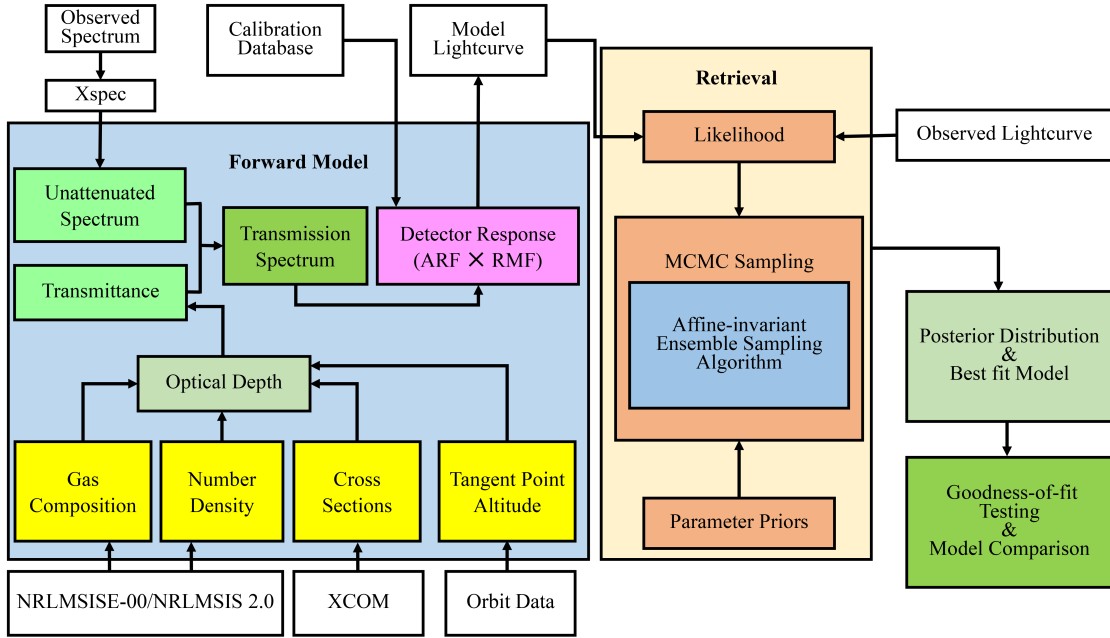

**Figure 6.** (Color online) The schematic of lightcurve modeling of the X-ray Earth occultation.

energy range of interest in this paper, the scattering effect can be ignored because it is too small relative to the photoelectric absorption effect. However, the scattering cross section is still included in the calculation.

The modelled lightcurves with this forward model are shown in Figure 7 from *Insight*-HXMT for the X-ray Earth occultation of the Crab Nebula. The normalized flux with the orbital phase is shown. From Figure 7, we can see that the occultation depths (the difference between the highest and lowest point of the same light curve) for different energy bands are very different. Here, the atmospheric model NRLMSISE-00 (Picone et al., 2002) is chosen as our input data in the lightcurve calculations in Figure 7.

$I_0$ was fitted by using Xspec, a standard software package for spectrum fitting in X-ray astronomy. To fit the unattenuated spectrum of the Crab Nebula, we use the model $wabs \times powerlaw$ (Godet et al., 2009; Yan et al., 2018), where $wabs$ is the interstellar absorption model in Xspec (Morrison and McCammon, 1983; Arnaud et al., 1999). The model $powerlaw$ represents a simple power law shape of spectrum to fit. The data description and fitting result of unattenuated energy spectrum are listed in Table 2. The reduced chi-squared (Mighell, 1999) is 1.06 in this fitting, which indicates that the fit is good. The best-fit model and the unattenuated energy spectrum data are shown in the upper panel of Figure 8. The blue dots with the error bars are the unattenuated spectrum of the Crab Nebula observed by LE. The red solid line is the best-fit model. The lower panel of Figure 8 shows the residuals of the fit.

The main atmospheric components causing extinction are Oxygen (O, $O_2$), Nitrogen (N, $N_2$) and Argon during the occultation. The absorption of N and O to X-ray has similar characteristics because the energy dependence of the X-ray cross sections

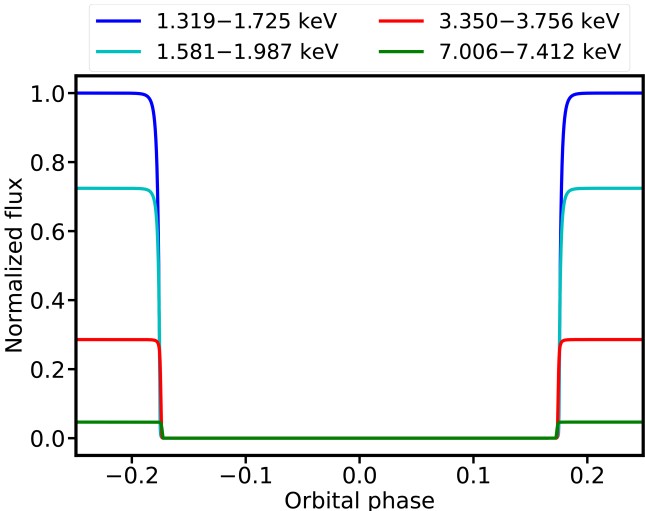

**Figure 7.** (Color online) The modelled lightcurves from *Insight*-HXMT for the X-ray Earth occultation of the Crab Nebula. The blue, cyan, red and green predicted lightcurves correspond to four different energy ranges, and these predicted lightcurves are all calculated based on the input data from the NRLMSISE-00 model. The normalized flux of each predicted lightcurve is represented with the orbital phase. It is found that the occultation depths (the difference between the highest and lowest point of the same light curve) of different energy segments are very different.

**Table 2.** Data description and fitting results of unattenuated energy spectrum.

| Data description | | Fitting parameters | | | Goodness of fit | |
| --- | --- | --- | --- | --- | --- | --- |
| Exposure time (s) | Energy range (keV) | $N_H$ | $K$ | $\alpha$ | $\chi^2/\mathrm{dof}$ | p-value |
| 99.75 | 1–10 | 0.3786 | 8.7822 | 2.1123 | 1.0613 | 0.3000 |

of N and O is similar in our interest energy band (Katsuda et al., 2021). In other words, it is impossible to distinguish N and O through X-ray occultation in our interest energy band, but their total atmospheric density (N+O+$O_2$+$N_2$) distribution can be calculated. In addition, although X-ray photons interact directly with the K- and L- shells electrons of atoms (including atoms within molecules), the $O_2$ (or $N_2$) counts as one absorbing "particles" in the calculation. Ar is an atmospheric constituent of less content relative to N and O in the Earth's atmosphere. The atmospheric density of Ar is 0.029%–0.943% of the total density of N and O according to the NRLMSISE-00 model in the altitude range of 20 km–200 km. But the X-ray cross section of Ar is larger by almost one order of magnitude than that of N and O over the energy range of interest. Therefore, Ar is included in our model.

The number density of each atmospheric component needs to be given as input data in the process of density profile retrieval with a forward model (Determan et al., 2007). In the following, the atmospheric model NRLMSISE-00 and NRLMSIS 2.0 (Emmert et al., 2021) are chosen as our input data in the modeling, respectively. The NRLMSISE-00 model is one of the

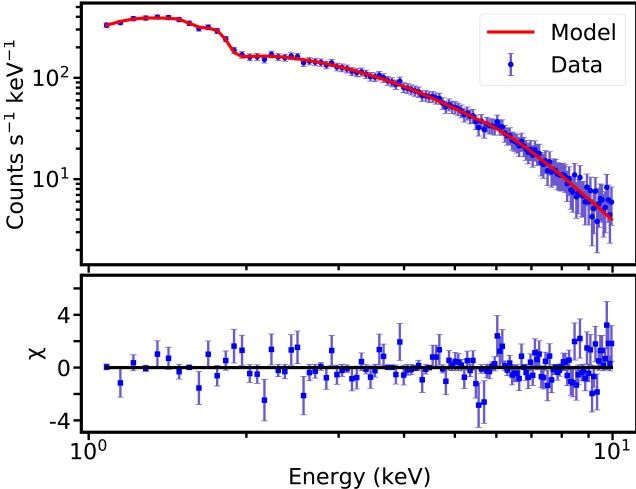

**Figure 8.** (Color online) Upper panel: The best-fit model and the data of unattenuated spectrum for the Crab Nebula from LE. The blue dots with error bars are observation data and the red solid line is the best-fit model. Lower panel: The residuals ((data-model)/error) of the fit.

most widely used and relatively accurate atmospheric model for the Earth's upper and middle atmosphere. The NRLMSIS 2.0 model is an upgraded version of the NRLMSISE-00 model. The input parameters of NRLMSISE-00/NRLMSIS 2.0 for the calculation of model atmospheric density are listed in Table 3. In Table 3, the time corresponds to the middle tangent point altitude during occultations. The geographical latitude and longitude are calculated with coordination tranformation from the coordination of the tangent point in J2000. $F_{10.7}$ and Ap are the solar activity index and the geomagnetic activity at this time. The $F_{10.7}$ index is one of the most widely used index to characterize the level of solar activity (Tapping, 2013), and the Ap index is used to characterize the geomagnetic activity (Clúa de Gonzalez et al., 1993). These data are obtained from the Space Environment Prediction Center [3].

The X-ray cross section $\sigma_s$ of each component of the gas can be obtained through the photon cross section database XCOM (Berger and Hubbell, 1987; Berger et al., 2010). The X-ray cross sections of Ar, O and N are shown in Figure 2.

**Table 3.** The input parameters of NRLMSISE-00 for the calculation of model atmospheric density profile.

| Latitude and longitude | Date and time (UTC) | $F_{10.7}$ (sfu) | $F_{10.7}$ average (sfu) | Ap (2nT) |
|---|---|---|---|---|
| $(57.05°, 71.39°)$ | 2018-09-30 15:39:22 | 68.0 | 68.7 | 6.0 |

The specific form of the forward model of X-ray occultations is as follows (Determan et al., 2007),

$$M = RI_0 e^{-\tau} + B, \tag{3}$$

---

[3] http://eng.sepc.ac.cn/index.php

where $R$ is the detector response matrix (Li et al., 2020). In addition, the forward model also contains background noise $B$. The background noise should be included in the forward model, as shown in eq. (3).

## 3.2 Density profile retrieval

Instead of subtracting the background, the background and the source counts can be modeled synchronously using Poisson statistics in the Bayesian framework (Olamaie et al., 2014). Bayes' theorem combines observation data with the prior distribution of the parameter of interest $\theta$ from a specific model to obtain the posterior probability distribution of the parameter. In this work, the posterior probability distribution of $\theta = \{\gamma, B\}$ for the forward model shown in eq. (3) applied to the data $D$ can be given by Bayes' theorem (Bayes and Price, 1763),

$$p(\theta|D, M) = \frac{p(\theta|M)p(D|\theta, M)}{p(D|M)}, \tag{4}$$

where $D$ is the observation data, $M$ is the forward model, $p(\theta|M)$ is the prior distribution, $p(D|\theta, M)$ is the likelihood and $p(D|M)$ is the Bayesian evidence.

Both the X-ray observed counts and the background data follow Poisson statistics so that the X-ray likelihood function, $\mathcal{L}$, is given by

$$\ln \mathcal{L} = \sum_i [D_i \ln M_i - M_i - \ln D_i!], \tag{5}$$

where $D_i$ is the observation data in the $i$th bin, $M_i$ is the $i$th forward model value. The natural logarithm of the likelihood function can also be used for parameter estimation as the C statistic (Cash, 1979).

In this paper, we analyze the lightcurve in the energy range of 1.0–2.5 keV, 2.5–6.0 keV and 6.0–10.0 keV, it is found that these energy ranges are indeed sensitive to the altitude range of 105–200 km, 95–125 km and 85–110 km, respectively, as shown in Figure 9. The red shadow in Figure 9 indicates the occultation range and the blue shadow in Figure 9 indicates the energy range.

The Markov Chain Monte Carlo (MCMC) method is one of the parameter estimation methods used for Bayesian inference (Sharma, 2017). The density profile retrieval implements the MCMC method, which samples from a probability distribution using Markov chains (Chib and Greenberg, 1995; Dunkley et al., 2005; Hogg and Foreman-Mackey, 2018). MCMC is implemented by `emcee` (Foreman-Mackey et al., 2013) that uses an affine-invariant ensemble sampler. A total of 200000 steps with 10 walkers are used in the sampling process. A 20000 step MCMC chain for each walker led to one standard deviation estimates for the correction factor $\gamma_s$ and the background $B$ as [0.871-0.905] and [0.609-0.720], [0.787-0.834] and [0.797-0.924], [0.811-0.948] and [1.399-1.558] based on NRLMSISE-00 and to one standard deviation estimates for the correction factor $\gamma_s$ and the background $B$ as [1.075-1.118] and [0.621-0.735], [0.897-0.951] and [0.793-0.919], [0.936-1.905] and [1.398-1.558] based on NRLMSIS 2.0 in the energy range of 1.0–2.5 keV, 2.5–6.0 keV and 6.0–10.0 keV, respectively, where the first 1000 steps in each walker are burned. The posterior probability distribution of the correction factor $\gamma_s$ and the background $B$ is obtained, as shown in Figure 10. In this corner plot, the vertical black dashed lines are the quantiles 0.16 and 0.84 of the posterior probability distribution respectively. The vertical red dashed line indicates the median for the posterior probability distribution

of the correction factor $\gamma_s$ and the background $B$. The median of the posterior probability distribution and the interval with the quantiles of 0.16 and 0.84 are marked on the top of the histogram. The retrieved results of atmospheric density can be obtained by multiplying $\gamma_s$ with the input data. Therefore, $\gamma_s$ can be used as an average scaling factor for NRLMSISE-00/NRLMSIS 2.0 density model.

The comparison for the retrieved density profile with XEOS in the energy range of 1.0–2.5 keV, 2.5–6.0 keV and 6.0–10.0 keV, the previous retrieval results based on RXTE in the altitude range of 100–120 km (Determan et al., 2007), the NRLMSISE-00 model density profile and the NRLMSIS 2.0 model density profile are shown in Figure 11. The density profiles marked by the legend of *Insight*-HXMT (XEOS-00) and *Insight*-HXMT (XEOS-2.0) in each panel in Figure 11 are the retrieved results based on the NRLMSISE-00 model and the NRLMSIS 2.0 model, respectively. There is an obvious gap between the XEO measurement and the model prediction from NRLMSISE-00. The retrieved density profiles from *Insight*-HXMT based on the NRLMSISE-00/NRLMSIS 2.0 models by fitting the lightcurves in the energy range of 1.0–2.5 keV, 2.5–6.0 keV and 6.0–10.0 keV are qualitatively consistent with the previous retrieved results from RXTE, and there are two intersections between the retrieved density with *Insight*-HXMT by fitting the lightcurve in the energy range of 2.5–6.0 keV and the retrieved density from RXTE. The XEO retrieved density is approximately 88.8% of the density of the NRLMSISE-00 model and 109.7% of the density of the NRLMSIS 2.0 model in the altitude range of 105–200 km at the same time and location, respectively. The XEO retrieved density is approximately 81.0% of the density of the NRLMSISE-00 model and approximately 92.3% of the density of the NRLMSIS 2.0 model in the altitude range of 95–125 km at the same time and location, respectively. The XEO retrieved density is approximately 87.7% of the density of the NRLMSISE-00 model and 101.4% of the density of the NRLMSIS 2.0 model in the altitude range of 85–110 km at the same time and location, respectively. The previous retrieval density based on RXTE by XOS is approximately 50% of the density of the NRLMSISE-00 model in the altitude range of 100–120 km on November 14, 2005. The latest semi-empirical atmospheric model, NRLMSIS 2.0, which belongs to the MSIS family, is recently released and it is a significant upgrade of the previous version, NRLMSISE-00. The density profile from NRLMSIS 2.0 is about 8.5%–21.9% lower than the densities from NRLMSISE-00 in the altitude range of 95–125 km. Nevertheless, the gap between the retrieved density profile and the model prediction from NRLMSISE-00 and NRLMSIS 2.0 still needs to be verified further.

Comparisons between the observed lightcurve data points and the model lightcurves based on the best-fit density profiles and model predicted lightcurves based on the NRLMSISE-00/NRLMSIS 2.0 model simulation are shown in Figure 12. Panels (a), (b) and (c) indicate the lightcurves in the energy range of 1.0–2.5 keV, 2.5–6.0 keV and 6.0–10.0 keV, respectively. The residuals between observed lightcurves and model lightcurves are also shown in Figure 12. For clarity, the data points for observed lightcurve in Figure 12 are displayed by taking one point every three points from the initial data points, and the residuals corresponding to the four lightcurves are shifted vertically in lower sub-panel of each panel. Four extinction curves of each panel are obtained by using the forward model based on the four density profiles in order to compare with the XEO observation data, among which the density profiles mainly include the two XEOS retrieved density profiles, the NRLMSISE-00/NRLMSIS 2.0 model predicted density profile at approximately the same date, time and geographic latitude and longitude.

In order to show the difference between the lightcurves, we amplified the observed lightcurve and the four model lightcurves in the altitude range of 120–124 km, 106–109 km and 95–100 km in panel (a), panel (b) and panel (c) in Figure 12, respectively.

## 3.3 Results testing

In this section, the Pearson's $\chi^2$ test (Pearson, 1900; Cochran, 1952) is used to test the XEO measurements, NRLMSISE-00/NRLMSIS 2.0 model prediction for the description of the XEO lightcurve.

In this study, the following null hypotheses are proposed. The lightcurves predicted by XEO measured density profile and NRLMSISE-00/NRLMSIS 2.0 model simulated density profile fit the observed lightcurve well, respectively. It is found that the null hypothesis can not be rejected even at 84%, 90% confidence level for the two XEO measurements in the energy range of 1.0–2.5 keV, the null hypothesis can not be rejected even at 55%, 64% confidence level for the two XEO measurements in the energy range of 2.5–6.0 keV and the null hypothesis can not be rejected even at 68%, 69% confidence level for the two XEO measurements in the energy range of 6.0–10.0 keV, as shown in Table 4. It is found that the null hypothesis can not be rejected at 95% confidence level for the NRLMSISE-00/NRLMSIS 2.0 predictions, respectively, except for the NRLMSISE-00 predictions by the model lightcurve in the energy range of 1.0–2.5 keV. Goodness-of-fit testing results between the observed lightcurve and the extinction curve predictions with XEO measured density profiles, the NRLMSISE-00/NRLMSIS 2.0 model simulated density profiles are also listed in Table 4.

Goodness-of-fit testing is carried out for the observed lightcurve and four model lightcurves in the energy range of 1.0–2.5 keV. As shown in Table 4, the $\chi^2/\mathrm{dof}$ and $p$-value between the observed lightcurve and the extinction curve predicted with XEO retrieved density profile based on NRLMSISE-00 are 1.0599 and 0.1604, where, dof represents degree of freedom, i.e., the number of sample points minus the number of variables. In this paper, 551 sample points are used for fitting, with two variables of correction factor and background noise, so dof=549. The $\chi^2/\mathrm{dof}$ and $p$-value between the observed lightcurve and the extinction curve predicted with XEO retrieved density profile based on NRLMSIS 2.0 are 1.0756 and 0.1074. The $\chi^2/\mathrm{dof}$ and $p$-value between the observed lightcurve and the extinction curve predicted with NRLMSISE-00 predicted density profile are 1.1220 and 0.0249. The $\chi^2/\mathrm{dof}$ and $p$-value between the observed lightcurve and the extinction curve predicted with NRLMSIS 2.0 predicted density profile are 1.0783 and 0.0997. The results show that the lightcurves based on XEO retrieved density can better describe the observed lightcurve. Compared with the retrieved results, the atmospheric density predicted by NRLMSISE-00 model overestimates by 11.2%, and the atmospheric density predicted by NRLMSIS 2.0 model underestimates by 9.7%.

Goodness-of-fit testing is carried out for the observed lightcurve and four model lightcurves in the energy range of 2.5–6.0 keV. As shown in Table 4, the $\chi^2/\mathrm{dof}$ and $p$-value between the observed lightcurve and the extinction curve predicted with XEO retrieved density profile based on NRLMSISE-00 are 1.0091 and 0.4540. The $\chi^2/\mathrm{dof}$ and $p$-value between the observed lightcurve and the extinction curve predicted with XEO retrieved density profile based on NRLMSIS 2.0 are 1.0612 and 0.3669. The $\chi^2/\mathrm{dof}$ and $p$-value between the observed lightcurve and the extinction curve predicted with NRLMSISE-00 predicted density profile are 1.3321 and 0.0802. The $\chi^2/\mathrm{dof}$ and $p$-value between the observed lightcurve and the extinction curve predicted with NRLMSIS 2.0 predicted density profile are 1.3331 and 0.0797. It is found that the lightcurve predictions

based on the XEO retrieved density profiles can describe the observed lightcurve better, and gaps between the retrieved density profiles and the model simulated ones exist. Compared with our retrieved results, the density profile of the NRLMSISE-00 model is overestimated by 19%, and the density profile of the NRLMSIS 2.0 model is overestimated by 7.7%.

Goodness-of-fit testing is carried out for the observed lightcurve and four model lightcurves in the energy range of 6.0–10.0 keV. As shown in Table 4, the $\chi^2/\mathrm{dof}$ and $p$-value between the observed lightcurve and the extinction curve predicted with XEO retrieved density profile based on NRLMSISE-00 are 1.0936 and 0.3293. The $\chi^2/\mathrm{dof}$ and $p$-value between the observed lightcurve and the extinction curve predicted with XEO retrieved density profile based on NRLMSIS 2.0 are 1.1012 and 0.3193. The $\chi^2/\mathrm{dof}$ and $p$-value between the observed lightcurve and the extinction curve predicter with NRLMSISE-00 predicted density profile are 1.2085 and 0.1967. The $\chi^2/\mathrm{dof}$ and $p$-value between the observed lightcurve and the extinction curve predicted with NRLMSIS 2.0 predicted density profile are 1.0955 and 0.3268. The results show that the lightcurves based on XEO retrieved density and the NRLMSIS 2.0 predicted density can better describe the observed lightcurve. The retrieved results based on the lightcurve in the energy range of 6.0–10.0 keV are basically consistent with the model density of NRLMSIS 2.0, but the density profile of NRLMSISE-00 is about 12.3% overestimated.

From the above results, it is found that the altitude ranges correspond to the lightcurve attenuations are indeed sensitive to different energy ranges and they often overlap. For example, the energy band of 1.0–2.5 keV is sensitive to the altitude range of 105–200 km, and the energy band of 6.0–10.0 keV is sensitive to the altitude range of 85–110 km. Therefore, the overlap range between the sensitive altitude ranges for the two energy bands is 105–110 km. But the retrieved density profiles of different energy bands are different in the overlap altitude range. This is because the XEOS retrieval by lightcurve fitting is an altitude-dependent method, different energy bands have different sensitive altitude ranges, and the retrieved results can be different even if there are overlapping altitude areas. The retrieved results are a simple scaling of MSIS density. The occultation data of X-ray detectors with larger effective area or stronger detection ability can be used to retrieve atmospheric density to reduce the influence of energy integration.

Since the Earth's middle and upper atmosphere is greatly affected by solar activity and geomagnetic activity, it will also have an impact on the density of the Earth's upper and middle atmosphere, so the possible systematic uncertainty of the predicted XEO lightcurve due to variations of solar and geomagnetic activity are discussed. Common solar activity includes sunspots, flares, corona, etc.. Here, the effects can be demonstrated by the predicted XEO lightcurve with NRLMSISE-00 density profile for the changing solar activities index $F_{10.7}$ and geomagnetic activities index Ap. The values of the model lightcurves under extreme solar activity and very low solar activity (Licata et al., 2020), a severe geomagnetic storm and quiet geomagnetic activity (Palacios et al., 2018) are calculated, respectively, as shown in Figure 13. The energy range of the lightcurve in panel (a), panel (b) and panel (c) in Figure 13 is 1.0–2.5 keV, 2.5–6.0 keV, and 6.0–10.0 keV, respectively. For clarity, the data points are displayed by taking one point every five points from the initial data points in Figure 13. In order to show the difference between the lightcurves under the different solar activities and geomagnetic activities, the observed lightcurve and the model lightcurves in the altitude range of 105–150 km, 95–125 km and 85–110 km are locally amplified in panel (a), panel (b) and panel (c) in Figure 13, respectively. The Akaike information criterion (AIC) and the Bayesian information criterion (BIC) are calculated for these model comparisons and the results are listed in Table 5. AIC and BIC are used for model selection, and

**Table 4.** Hypothesis testing results for the extinction curve predictions with XEO measured density profiles, NRLMSISE-00/NRLMSIS 2.0 model simulated density profiles (during the occultation).

| Energy | Method | $\chi^2/\mathrm{dof}$ | $p$-value |
|---|---|---|---|
| 1.0–2.5 keV | XEOS-00 | 1.0599 | 0.1604 |
| | XEOS-2.0 | 1.0756 | 0.1074 |
| | NRLMSISE-00 | 1.1220 | 0.0249 |
| | NRLMSIS 2.0 | 1.0783 | 0.0997 |
| 2.5–6.0 keV | XEOS-00 | 1.0091 | 0.4540 |
| | XEOS-2.0 | 1.0612 | 0.3669 |
| | NRLMSISE-00 | 1.3321 | 0.0802 |
| | NRLMSIS 2.0 | 1.3331 | 0.0797 |
| 6.0–10.0 keV | XEOS-00 | 1.0936 | 0.3293 |
| | XEOS-2.0 | 1.1012 | 0.3193 |
| | NRLMSISE-00 | 1.2085 | 0.1967 |
| | NRLMSIS 2.0 | 1.0955 | 0.3268 |

can also be used to compare models. Usually, we choose the model with the smallest AIC and BIC. However, the values of AIC and BIC in this paper show that solar and geomagnetic activity has a great influence on model shape. Because AIC and BIC values vary greatly under different solar and geomagnetic activity in a relatively low energy range. Goodness of fit between the observed lightcurve and the model lightcurves under the different solar activities and geomagnetic activities is also evaluated by $\chi^2/\mathrm{dof}$ and $p$-value in Table 5. It can be shown that the solar activity and the geomagnetic activity have great influence on the shape of model lightcurves. In addition, with the increase of altitude, solar and geomagnetic activities have a greater impact on the model lightcurves. The effects of solar activities and geomagnetic storms for the XEO lightcurve modeling and density retrieval will be further investigated in the future.

### 3.4 Comparison to the results from altitude independent method by spectrum fitting

Based on the energy spectrum fitting method during X-ray occultation (Katsuda et al., 2021; Yu et al., 2022), the altitude independent atmospheric density retrieved results can be obtained, and the overlap of the tangent point altitude can be effectively avoided. In order to prove the reliability of our retrieved results in the paper, we compare our results with the results of energy spectrum fitting (Yu et al., 2022). The optical depth of the forward model based on energy spectrum fitting is given by the following equation,

$$\tau = \sum_s \gamma_h \int_{l_0}^{\infty} n_s \sigma_s dl, \tag{6}$$

**Table 5.** The calculated values of AIC, BIC, $\chi^2/\mathrm{dof}$ and $p$-value.

| Energy | Model | AIC | BIC | $\chi^2/\mathrm{dof}$ | $p$-value |
|---|---|---|---|---|---|
| | Extreme solar activity | 347.7317 | 358.3515 | 3.0768 | 0.0 |
| | Very low solar activity | 153.6371 | 164.2570 | 1.2890 | 0.0177 |
| 1.0–2.5 keV | Severe geomagnetic storm | 704.2318 | 714.8516 | 6.2610 | 0.0 |
| | Quiet geomagnetic activity | 181.7213 | 192.3412 | 1.5636 | $7.3075 \times 10^{-5}$ |
| | Extreme solar activity | 50.3180 | 60.9379 | 1.2735 | 0.1177 |
| | Very low solar activity | 53.2962 | 63.9160 | 1.3831 | 0.0562 |
| 2.5–6.0 keV | Severe geomagnetic storm | 152.6819 | 163.3017 | 3.4441 | $2.1303 \times 10^{-12}$ |
| | Quiet geomagnetic activity | 71.0399 | 81.6597 | 1.9109 | 0.0005 |
| | Extreme solar activity | 40.4108 | 51.0307 | 1.1069 | 0.3118 |
| | Very low solar activity | 40.9783 | 51.5981 | 1.1646 | 0.2422 |
| 6.0–10.0 keV | Severe geomagnetic storm | 42.8781 | 53.4980 | 1.0945 | 0.3282 |
| | Quiet geomagnetic activity | 41.4147 | 52.0346 | 1.1842 | 0.2211 |

where $\gamma_h$ represents correction factors in different altitudes ranges. By combining Eq. (3) and Eq. (6), the forward model of the energy spectrum fitting for different altitude ranges is given. By fitting the energy spectrum data in different altitude ranges, the correction factors in corresponding altitude ranges can be obtained, namely $\gamma_h$. So multiplying $\gamma_h$ with the input data from the NRLMSIS 2.0 model, the atmospheric density in different altitude ranges can be retrieved independently.

In the paper, by fitting the energy spectrum data in the energy range of 1–10 keV, we obtain the atmospheric density values in the altitude range of 100–200 km, and extract the energy spectrum data every 10 km. The comparison between the best-fit model and energy spectrum observational data is shown in Figure 14. The retrieved results based on energy spectrum fitting and the results with lightcurve fitting are shown in Figure 15, where the solid blue line represents the retrieved results of spectrum fitting, the solid red line represents the model density profile of NRLMSIS 2.0, and the solid green line represents the retrieved results with lightcurve fitting in the energy range of 1.0–2.5 keV. It is found that the retrieved results based on the lightcurve fitting are qualitatively consistent with the retrieved results of the energy spectrum fitting method. In the altitude range of 180–200 km, because the number of X-ray photons absorbed by the Earth's atmosphere is less than the X-ray photon counting error, the retrieved results based on energy spectrum fitting have large uncertainty. However, the reliability of the results based on lightcurve fitting is proved to be consistent with that by altitude independent method by spectrum fitting.

## 4 Conclusions and discussions

In this paper, we have studied the X-ray Earth occultation of the Crab Nebula with the pointing observation data from *Insight*-HXMT. We have presented a detailed Bayesian data analysis method for the extinction lightcurve modeling from the X-ray Earth occultation process. The theoretical predicted XEO observational lightcurve is calculated with the lightcurve forward

model. The data recorded by the Low Energy X-ray telescope (LE) of *Insight*-HXMT are analyzed and the density profile is retrieved. The results are tested and validated with the measurements from the RXTE satellite and the retrieval results with the altitude independent method by spectrum fitting.

We have shown from the XEO extinction lightcurve modeling that the X-ray astronomical satellite *Insight*-HXMT can be used to retrieve atmospheric density by the X-ray Earth occultation of celestial sources. The XEO retrieved density profile in the altitude range of 105–200 km by fitting the lightcurves in the energy range of 1.0–2.5 keV is lower than the density of NRLMSISE-00 and larger than the density of NRLMSIS 2.0 at the same date, time and geographical location. The results show that the retrieved density profiles are 88.8% of the NRLMSISE-00 density and 109.7% of the NRLMSIS 2.0 density in the altitude range of 105–200 km. The XEO retrieved density profile in the altitude range of 95–125 km by fitting the lightcurves in the energy range of 2.5–6.0 keV is lower than the NRLMSISE-00/NRLMSIS 2.0 model predicted density profile at the same date, time and geographical location. It is found that the retrieved density profile is 81.0% of the density prediction by NRLMSISE-00 and the retrieved density profile is 92.3% of the density prediction by NRLMSIS 2.0 in an altitude range of 95–125 km, respectively. The XEO retrieved density profile in the altitude range of 85–110 km by fitting the lightcurves in the energy range of 6.0–10.0 keV is lower than the NRLMSISE-00 density, but almost consistent with the density of the NRLMSIS 2.0 model. Moreover, the XEO retrieved density profiles and the NRLMSIS 2.0 density can better describe the lightcurve than the density of NRLMSISE-00 at the same time, date and location. The results show that the XEO retrieved density profiles are 87.7% of the NRLMSISE-00 density and 101.4% of the NRLMSIS 2.0 density. The retrieved density profiles with LE/*Insight*-HXMT are qualitatively consistent with the retrieved density with RXTE, especially in the altitude range of 95–125 km, there are two intersections between the XEO retrieved density and the measurement of RXTE. This shows that the *Insight*- HXMT, as an atmospheric diagnostic instrument, further validates the difference between the measurements from the X-ray satellite and the model density. The *Insight*-HXMT satellite with other X-ray astronomical satellites in orbit can form a space observation network for XEOS in the future[4]. In addition, it is found that the sensitive altitude ranges of different energy bands often overlap, and the retrieved atmospheric densities of the overlapping regions obtained by fitting the lightcurves of different energy ranges are often different. The lightcuve fitting method is the altitude-dependent method, different energy bands have different sensitive altitude ranges, the retrieved results are different even if there is an overlapping altitude range. In other words, an averaged scale factor for density profile in an altitude range is obtained by the lightcurve fitting method. We confirmed the measured density profile from lightcurve fitting by comparing to the ones by a standard spectrum retrieval method with an iterative inversion technique. The occultation data from larger effective areas can be used to retrieve atmospheric density to reduce the influence of energy integration. A more detailed description of this problem will be discussed in the future.

The difference between the measured and model values may result from the long-term accumulation of greenhouse gases, imperfect climatological estimates of solar and geomagnetic effects, differences in temperature profiles, and the influence of gravity waves (Determan et al., 2007; Katsuda et al., 2021). And the differences can also be due to model errors and/or retrieval errors. In order to further clarify and explain the difference, a large amount of X-ray occultation data from past, present, and

---

[4]https://heasarc.gsfc.nasa.gov/

future X-ray satellites will be required for further analysis. However, the atmospheric density retrieval method in this study depends on atmospheric models. The retrieved results vary with the shape of density profiles for different atmospheric models. Therefore, model independent retrieval methods needs to be developed, and we will consider this kind of XEOS method in the future.

X-ray photons with higher energy can penetrate deeper into the Earth's atmosphere. The observation data of HE and ME (Liu et al., 2020; Cao et al., 2020) will be analyzed in our future work. In addition, we will investigate the factors for affecting the XEO lightcurve modeling and density retrieval, such as the extended X-ray source effects, the energy spectra variations, etc.. These effects for the the XEO lightcurve modeling and density retrieval will be analyzed in the next work.

*Data availability.* Datasets related to the paper are available from http://archive.hxmt.cn/proposal

*Author contributions.* BL and HL were responsible for conceptualisation. HL was responsible for funding acquisition and supervision. HL was responsible for the forward model building, the retrieval algorithm and the original software development. DY and YT were responsible for the data reduction. DY, HL and YL were involved in the software update and data analysis. HL and MG were responsible for the design of the observations. XL and WX were responsible for the validation of the results. DY prepared the original draft. All co-authors reviewed and edited the paper.

*Competing interests.* The authors declare that no competing interests are present.

*Acknowledgements.* HL thanks Dr. Viktoria Sofieva (Finnish Meteorological Institute), Dr. Xiaocheng Wu (National Space Science Center, CAS) for discussion and help on the treatment of Abel integral. This work was supported by the Youth Innovation Promotion Association CAS (Grant No. 2018178), the National Natural Science Foundation of China (Grant Nos. 41604152, U1938111, U1938109, U1838104, U1838105), the Strategic Priority Research Program on Space Science of Chinese Academy of Sciences (Grant Nos. XDA04060900, XDA15020800, XDA15072103), the National Key Research and Development Program of China (Grant Nos. 2017YFB0503300, 2016YFA0400800). This work made use of the data from the HXMT mission, a project funded by China National Space Administration (CNSA) and the Chinese Academy of Sciences (CAS). We greatly appreciate the anonymous referees for the insightful comments that helped improve the quality of this work.

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

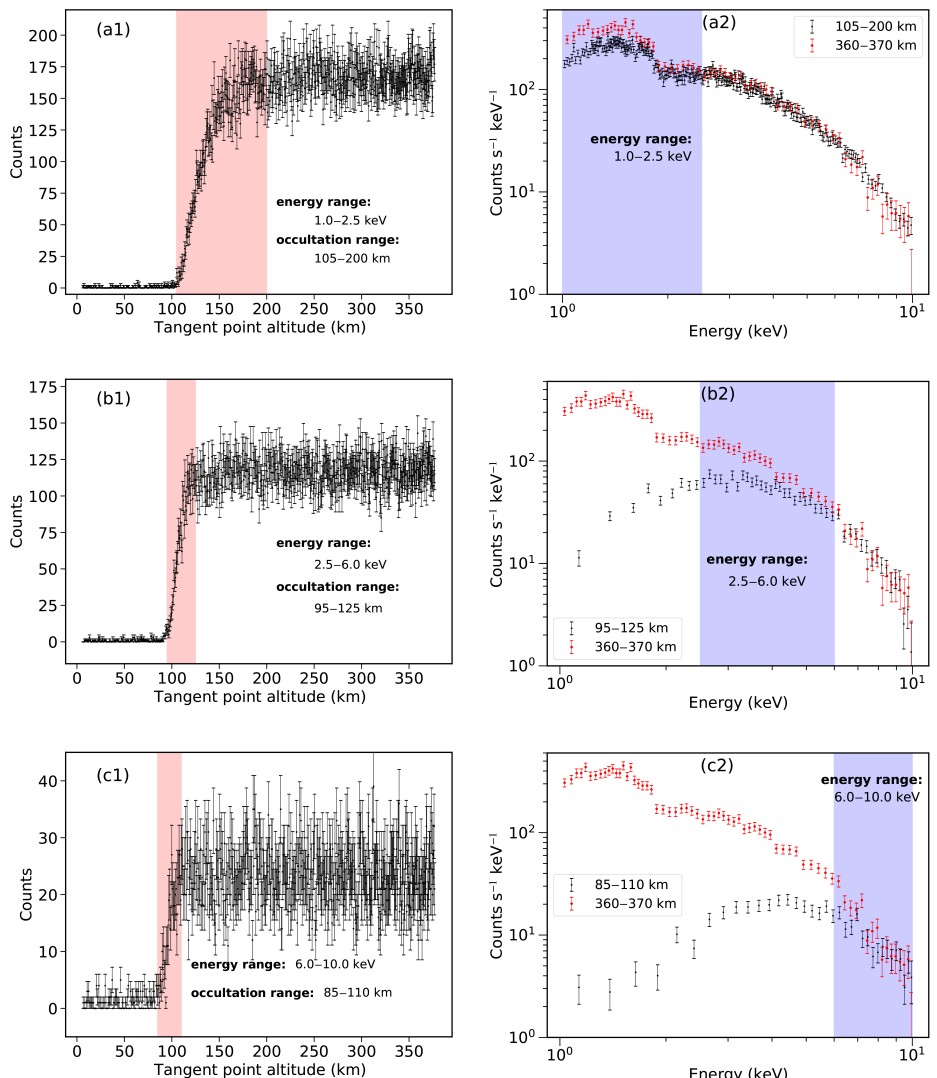

**Figure 9.** (Color online) Lightcurve and energy spectrum during occultation. Panel (a1) represents the lightcurve in the energy range of 1.0–2.5 keV, and it is found that this energy band is indeed sensitive to the altitude range of 105–200 km. Panel (a2) shows the comparison between the energy spectrum in the altitude range of 105–200 km and the unattenuated energy spectrum, and it is found that there is a significant attenuation of the energy spectrum in the altitude range in the range of 1.0–2.5 keV. Panel (b1) represents the lightcurve in the energy range of 2.5–6.0 keV, and it is found that this energy band is indeed sensitive to the altitude range of 95–125 km. Panel (b2) shows the comparison between the energy spectrum in this altitude range and the unattenuated energy spectrum, and it is found that the energy spectrum in this altitude range has significant attenuation in the energy range of 2.5–6.0 keV. Panel (c1) represents the lightcurve in the energy range of 6.0–10.0 keV, which is indeed sensitive to the altitude range of 85–110 km. Panel (c2) shows the comparison between the energy spectrum in this altitude range and the unattenuated energy spectrum, and it is found that the energy spectrum in this altitude range has significant attenuation in the energy range of 6.0–10.0 keV. The red shadow marks the occultation range and the blue shadow marks the energy range.

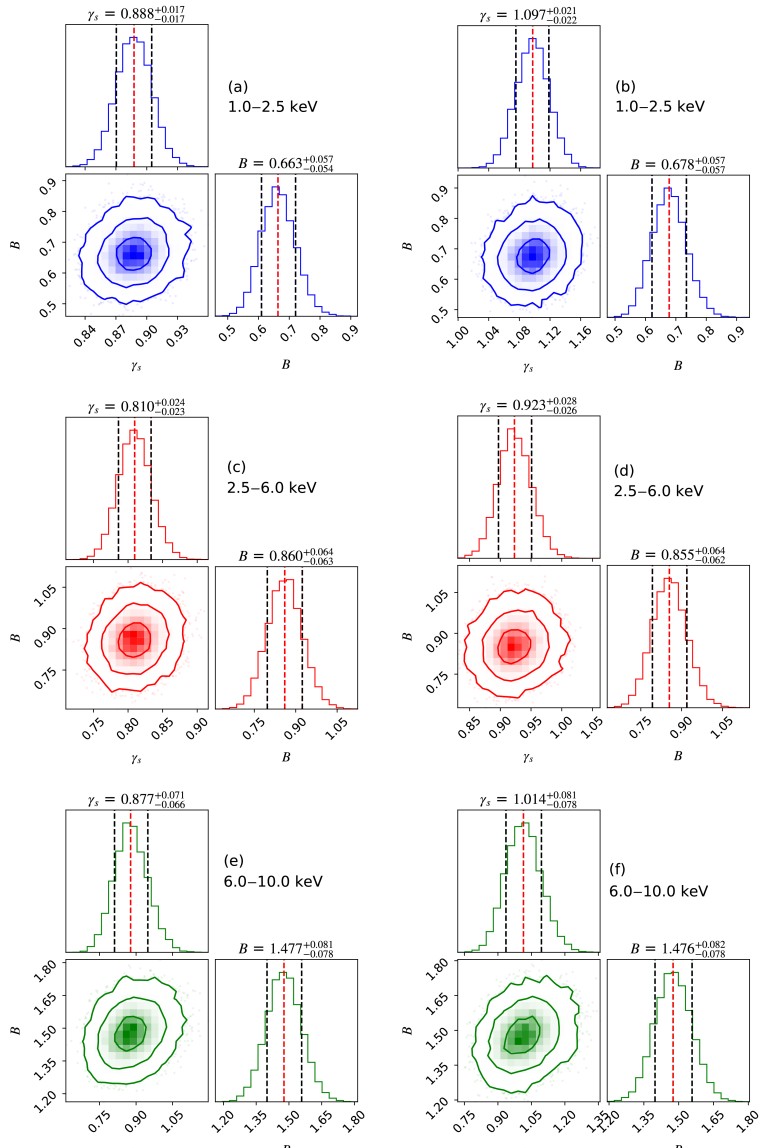

**Figure 10.** (Color online) Corner plot showing the one and two dimensional projections of the posterior probability distributions for the two free parameters. On the 2D plots, blue, red and green contours represent 1-, 2- and 3-$\sigma$ confidence intervals. The black dashed lines in each of the 1D histograms represent the quantiles 0.16 and 0.84 of the posterior probability distribution with the central red dashed line indicating the median value. The median $\pm$ the 68% confidence interval is shown above each histogram. Panels (a) and (b) represent posterior probability distributions based on NRLMSISE-00 and NRLMSIS 2.0, respectively, by fitting the lightcurves in 1.0–2.5 keV. Panel (c) and (d) represent posterior probability distributions based on NRLMSISE-00 and NRLMSIS 2.0, respectively, by fitting the lightcurves in 2.5–6.0 keV. Panel (e) and (f) represent posterior probability distributions based on NRLMSISE-00 and NRLMSIS 2.0, respectively, by fitting the lightcurves in 6.0–10.0 keV.

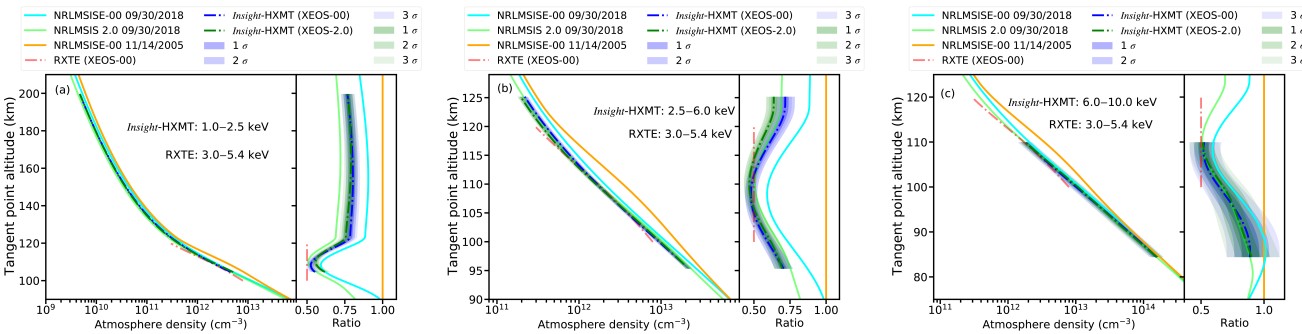

**Figure 11.** (Color online) Panel (a) shows the comparison between the retrieved density profile in the altitude range of 105–200 km based on XEOS method and the NRLMSISE-00/NRLMSIS 2.0 model predictions as well as the the previous retrieved results based on PCA/RXTE. Panel (b) shows the comparison between the retrieved density profile in the altitude range of 95–125 km based on XEOS method and the NRLMSISE-00/NRLMSIS 2.0 model predictions as well as the the previous retrieved results based on PCA/RXTE. Panel (c) shows the comparison between the retrieved density profile in the altitude range of 85–110 km based on XEOS method and the NRLMSISE-00/NRLMSIS 2.0 model predictions as well as the the previous retrieved results based on PCA/RXTE. Right sub-panel of each panel: all density profiles are normalized to the NRLMSISE-00 density profile on November 14, 2005, in order to visually compare the differences between the various density profiles.

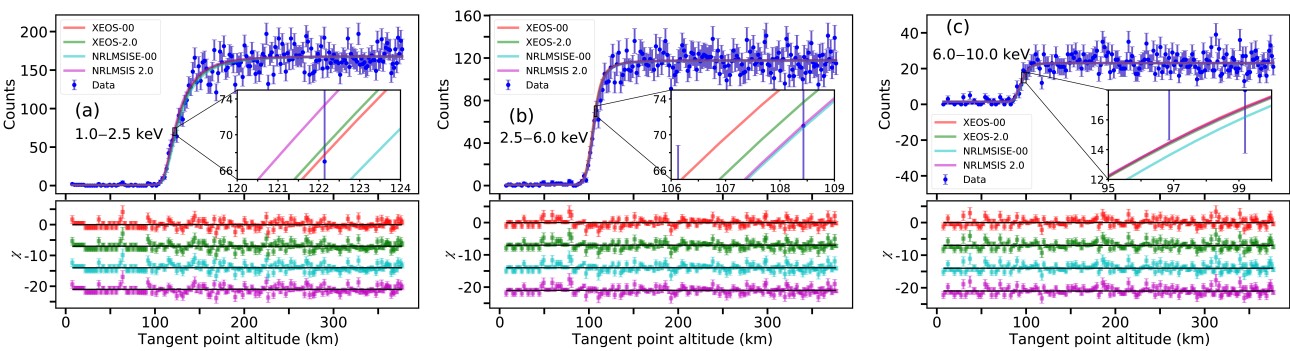

**Figure 12.** (Color online) Panel (a) represents the comparison between the observed lightcurve and best-fit lightcurves from XEOS measurement based on NRLMSISE-00 and NRLMSIS 2.0 as well as the model lightcurve based on the NRLMSISE-00/NRLMSIS 2.0 model predictions in the energy range of 1.0–2.5 keV. Panel (b) represents the comparison between the observed lightcurve and best-fit lightcurves from XEOS measurement based on NRLMSISE-00 and NRLMSIS 2.0 as well as the model lightcurve based on the NRLMSISE-00/NRLMSIS 2.0 model predictions in the energy range of 2.5–6.0 keV. Panel (c) represents the comparison between the observed lightcurve and best-fit lightcurves from XEOS measurement based on NRLMSISE-00 and NRLMSIS 2.0 as well as the model lightcurve based on the NRLMSISE-00/NRLMSIS 2.0 model predictions in the energy range of 6.0–10.0 keV. Lower sub-panel of each panel: the residuals between the observed lightcurve and best-fit lightcurves, model lightcurves based on NRLMSISE-00/NRLMSIS 2.0. The color of residuals corresponds to the color of the best-fit lightcurves or model lightcurves. The residuals corresponding to the four lightcurves are shifted vertically for clarity.

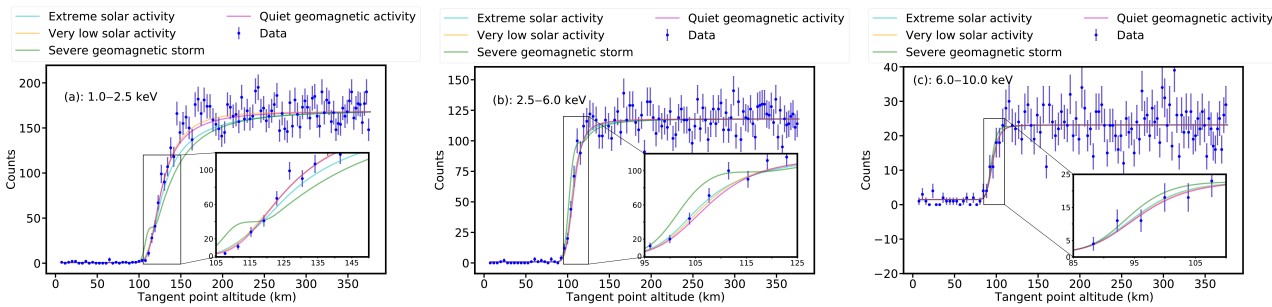

**Figure 13.** (Color online) Comparison of the observed data and forward model lightcurves under extreme solar activity, very low solar activity, severe geomagnetic storm, quiet geomagnetic activity. The NRLMSISE-00 density profiles are used as input data for those simulations. For clarity, the data points are displayed by taking one point every five points from the initial data points. The energy range of the lightcurves in panel (a), (b) and (c) is 1–2.5 keV, 2.5–6.0 keV and 6.0–10.0 keV, respectively. Furthermore, local magnification of the lightcurves in the altitude range of 105–150 km, 95–125 km and 85–110 km is carried out to show the influence of solar and geomagnetic activities on the shape of the model lightcurves in panel (a), (b) and (c), respectively.

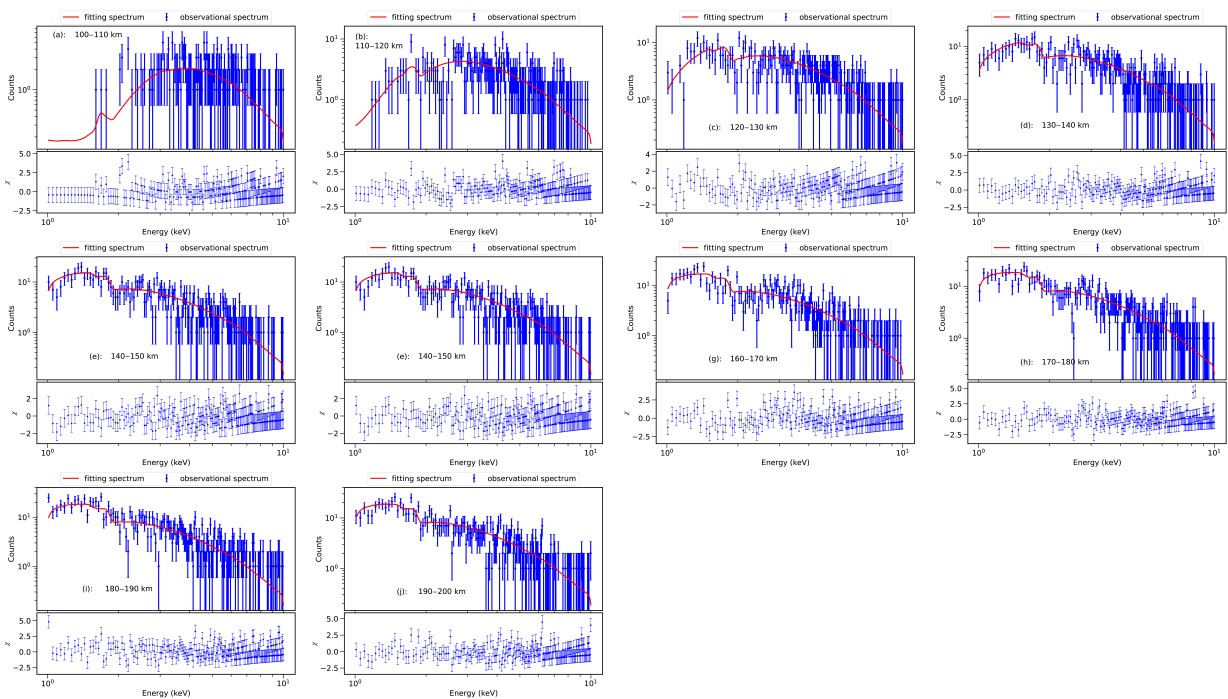

**Figure 14.** (Color online) Comparison of best fit spectrum model and observational spectrum data. In the upper space of each panel, blue dots with error bars represent data points, solid red lines represent best fit spectrum models, and the lower space of each panel represents residuals between the best fit model and observational spectrum data.

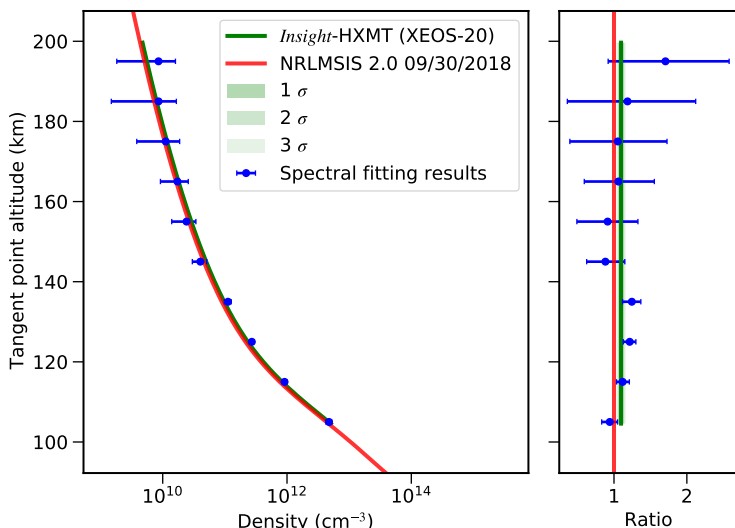

**Figure 15.** (Color online) Comparison of retrieved results based on energy spectrum fitting and lightcurve fitting. The solid blue line represents the retrieved results of spectrum fitting, the solid red line represents the model density profile of NRLMSIS 2.0, and the solid green line represents the retrieved results of lightcurve in the energy range of 1.0–2.5 keV. Right panel: all density profiles are normalized to the NRLMSISE 2.0 density profile on September 30, 2018.