# Peer review of "Measurement of the vertical atmospheric density profile from the X-ray Earth occultation of the Crab Nebula with *Insight*-HXMT"

_Atmospheric Measurement Techniques, 2021_

## Author Comment (AC1)

**In reference to AMT-2021-406 "Measurement of vertical atmospheric density profile from the X-ray Earth occultation of the Crab Nebula with *Insight*-HXMT":**

The authors are very grateful to the referees for their valuable comments and suggestions. Our responses to these comments are as follows.

Comments from reviewers:

**Reviewer #2:**

General comments:

1. **This paper presents a new measurement of the atmospheric density, based on the atmospheric occultation of X-ray emission from the Crab Nebula observed with *Insight*-HXMT. The authors analyzed a single egress event occurred on 2018-09-30T15:38:36. They showed that the density in altitude range of 105-200 km, 95-125 km, and 85-110 km are 88.8% (109.7%), 81.0% (92.3%), and 87.7% (101.4%) of the density prediction by NRLMSISE-00 (NRLMSIS 2.0), respectively. The density is qualitatively consistent with the previous results with RXTE. This study clearly demonstrates that *Insight*-HXMT can provide an approach for the study of the occultation sounding of the upper atmosphere.**

**Reply:** Thanks to the reviewer's comprehensive summary of the work and inspiring comments. The manuscript has been revised carefully based on the received comments. For details, please see the following responses.

Specific comments:

1. **I suggest the authors to estimate the uncertainty on the tangent point altitude. Two main sources of altitude errors are satellite position and pointing direction. Currently, the authors assume that these two parameters are perfectly known. However, it would be better to quantitatively give their errors and estimate how the errors affect the tangent altitude.**

**Reply:** Thank you very much for your valuable comments. We first analyzed the tangent point altitude errors from the satellite position. Monte Carlo simulation combined with satellite position measurement errors was used to give the uncertainties of the tangent point altitudes. The total position deviation of *Insight*-HXMT is 100 m under $3\sigma$ (3 standard deviation). We conducted 100 independent Gaussian sampling in the x and y directions of the satellite with $\mu = 0$ and $\sigma_x = \sigma/3$ ($\sigma_y = \sigma/3$). The standard deviation of satellite in the z direction is obtained by Eq. (1). The Gaussian noises were added to each direction of the satellite positions. Based on the satellite positions after adding noises, we calculated the tangent point altitudes. We analyzed the altitude errors caused by satellite position measurement errors and also estimate the density fitting errors caused by

uncertainties of the tangent point altitudes. The noise in one sample from one hundred independent Gaussian samples in the three directions is shown in FIG. 1 in this Response, where each sample in each direction has 552 points corresponding to 552 tangent point altitudes. Adding the corresponding noises in each direction of the satellite positions, we obtain the simulated positions of the satellite, as shown in FIG. 2 in this Response, which are the foundation for calculating the tangent point altitudes.

$$\sigma_x^2 + \sigma_y^2 + \sigma_z^2 = \sigma^2 \tag{1}$$

[Figure]

FIG. 1 in this Response: The noise in one sample from one hundred independent Gaussian samples in the three directions.

[Figure]

FIG. 2 in this Response: The simulated satellite positions after adding Gaussian noise to each direction of the satellite position.

The corresponding tangent point altitudes were obtained by analyzing the satellite position after adding noise, as shown in FIG. 3 in this Response. It was found that the error of the satellite's position resulted in a maximum deviation of about 40 m for the test tangent point altitude and the real altitude in the original manuscript, as shown the lower panel in FIG 3. in this Response.

[Figure]

FIG. 3 in this Response: The comparison between the simulated tangent altitude and the real altitudes (upper panel).The difference between the simulated tangent altitude and the real altitudes (lower panel).

The atmospheric density was fitted based on the simulated altitudes with error, and the value of correction factor γ was obtained. Since we had taken one hundred independent samples of Gaussian noise for the satellite position, we can get one hundred γ values.

Table 1 in this Response. The fitted one hundred γ values based on one hundred simulated tangent altitudes with error.

| Number | 1 | 2 | 3 | 4 | 5 | 6 | 7 | 8 | 9 | 10 |
|---|---|---|---|---|---|---|---|---|---|---|
| γ | 0.886361 | 0.886189 | 0.889484 | 0.887557 | 0.887338 | 0.887903 | 0.888294 | 0.885130 | 0.887975 | 0.888256 |
| Number | 11 | 12 | 13 | 14 | 15 | 16 | 17 | 18 | 19 | 20 |
| γ | 0.887792 | 0.886524 | 0.886981 | 0.888782 | 0.886298 | 0.886432 | 0.885342 | 0.887291 | 0.885377 | 0.886704 |

| Number | 21 | 22 | 23 | 24 | 25 | 26 | 27 | 28 | 29 | 30 |
|---|---|---|---|---|---|---|---|---|---|---|
| γ | 0.887477 | 0.887847 | 0.889383 | 0.885718 | 0.888707 | 0.886319 | 0.886040 | 0.889365 | 0.888271 | 0.890716 |

| Number | 31 | 32 | 33 | 34 | 35 | 36 | 37 | 38 | 39 | 40 |
|---|---|---|---|---|---|---|---|---|---|---|
| γ | 0.886064 | 0.889049 | 0.887571 | 0.887625 | 0.887682 | 0.888805 | 0.887769 | 0.886466 | 0.887387 | 0.886382 |

| Number | 41 | 42 | 43 | 44 | 45 | 46 | 47 | 48 | 49 | 50 |
|---|---|---|---|---|---|---|---|---|---|---|
| γ | 0.888116 | 0.887688 | 0.887036 | 0.885247 | 0.888977 | 0.886960 | 0.889081 | 0.890935 | 0.887302 | 0.888472 |

| Number | 51 | 52 | 53 | 54 | 55 | 56 | 57 | 58 | 59 | 60 |
|---|---|---|---|---|---|---|---|---|---|---|
| γ | 0.885955 | 0.888675 | 0.886698 | 0.887132 | 0.886116 | 0.888758 | 0.885437 | 0.886343 | 0.889651 | 0.884448 |

| Number | 61 | 62 | 63 | 64 | 65 | 66 | 67 | 68 | 69 | 70 |
|---|---|---|---|---|---|---|---|---|---|---|
| γ | 0.885307 | 0.887394 | 0.888518 | 0.890139 | 0.887470 | 0.886591 | 0.887623 | 0.887347 | 0.888659 | 0.887169 |

| Number | 71 | 72 | 73 | 74 | 75 | 76 | 77 | 78 | 79 | 80 |
|---|---|---|---|---|---|---|---|---|---|---|
| γ | 0.887695 | 0.888246 | 0.885270 | 0.888409 | 0.887954 | 0.889285 | 0.886362 | 0.887950 | 0.887805 | 0.888645 |

| Number | 81 | 82 | 83 | 84 | 85 | 86 | 87 | 88 | 89 | 90 |
|---|---|---|---|---|---|---|---|---|---|---|
| γ | 0.889220 | 0.887716 | 0.885728 | 0.886952 | 0.885945 | 0.886264 | 0.886824 | 0.887072 | 0.887083 | 0.888430 |

| Number | 91 | 92 | 93 | 94 | 95 | 96 | 97 | 98 | 99 | 100 |
|---|---|---|---|---|---|---|---|---|---|---|
| γ | 0.888386 | 0.887020 | 0.888858 | 0.888251 | 0.888904 | 0.885684 | 0.886363 | 0.886484 | 0.887122 | 0.886926 |

To show more clearly the relationship between the one hundred γ values and the γ values based on real altitude, we plotted the distribution of the one hundred sample results, as shown in FIG.4 in this Response. It is found that the error of satellite position has little

influence on the final retrieved results (the retrieved density results are all within the real density value $\pm 1\sigma$).

[Figure]

FIG. 4 in this Response: The values corresponding to the solid red line represent the retrieved results of γ by fitting the light curve in the energy range of 1.0-2.5 keV, the solid green line represents the real value $\pm 1$ σ. The shaded blue area shows the distribution of one hundred γ sample values. It is found that the error of satellite position has little influence on the final retrieved results

[Figure]

FIG. 5 in this Response: Observe geometry. The tangent height deviation ( △h) caused by the position error of the source (1 mas).

Then we analyzed the tangent point altitudes error caused by the source position error. If the position of the source is offset by 1 milliarcsecond, then the tangent height deviation ( △h) can be calculated. Through calculation, it is found that the tangent point altitude error caused by the position error of the source is on the order of centimetres or

millimetres, so the tangent point altitude error caused by the source position error can be ignored, because the tangent point altitude error is too small. As shown in FIG. 6 in this Response.

[Figure]

FIG. 6 in this Response: Tangent height error ($\Delta$ h) due to source error (1 milliarc second).

There are two main factors that can affect the tangent point altitudes according to our calculation process of the tangent point altitudes, namely, the position of the satellite and the position of the observation source. Through the above calculation, we have obtained the tangent point altitude error caused by the satellite position error and the tangent point error caused by the source position error. It is found that the tangent point altitude error caused by satellite position error and source position error can be ignored because it has little influence on the final retrieved density.

In addition, you mentioned the influence of satellite pointing direction on the tangent point altitudes error. The precision of pointing of *Insight*-HXMT: ≤0.1 degree (3σ), the precision of measurement: ≤0.01 degree (3σ), and the stability of pointing: ≤ 0.005 degree/s (3σ). We think that the pointing direction will not affect the calculation of the tangent altitudes. *Insight*-HXMT is a collimated satellite, the field of view of LE detectors is 1.6 °*6 °. We correct the response of the collimator effect in the response matrix file when the incident direction of the photon is not perpendicular to the detector plane. The satellite may have jitter when it points to a required source, and this will lead to the change of the source direction to the detector plane and also the detected rate of the photons. This effect is considered in the response matrix file.

**2. Table 5 and Figure 13: It may be interesting to add the light curve in 1.0-2.5 keV, because the lower-energy band (i.e., higher altitude) seems more sensitive to the solar activity. Also, it would be better to bin the data (rather than sub-sampling as the authors did in the current paper) to improve the photon statistics.**

**Reply:** Thank you very much for your valuable suggestions. According to your suggestion, we added the lightcurves in the energy range of 1.0–2.5 keV, because the lower energy range corresponds to a higher tangent point altitude during occultation, and the higher altitude is more sensitive to solar activity, as shown in panel (a) in FIG. 7 in this Response. In addition, we also added the lightcurve in the energy range of 6.0–10.0 keV to compare the influence of solar and geomagnetic activities on different altitudes, as shown in panel (c) in FIG. 6 in this Response. The Akaike information criterion (AIC) and the Bayesian information criterion (BIC) are calculated for these model comparisons and the results are listed in Table 2 in this Response. Goodness of fit between the observed lightcurve and the model lightcurves under the different solar activities and geomagnetic activities is also evaluated by $\chi^2/$dof and $p$-value in Table 1 in this Response. It is found that the solar activity and the geomagnetic activity have great influence on the shape of model lightcurves. In addition, with the increase of altitude, solar and geomagnetic activities have a greater impact on the model lightcurves.

[Figure]

FIG. 7 in this Response: Comparison of the observed data and forward model lightcurves under extreme solar activity, very low solar activity, severe geomagnetic storm, quiet geomagnetic activity. For clarity, the data points are displayed by taking one point every five points from the initial data points. The energy range of the lightcurves in panel (a), (b) and (c) is 1–2.5 keV, 2.5–6.0 keV and 6.0–10.0 keV, respectively. Furthermore, local magnification of the lightcurves in the altitude range of 105–150 km, 95–125 km and 85–110 km is carried out to show the influence of solar and geomagnetic activities on the shape of the model lightcurves in panel (a), (b) and (c), respectively.

Table 2 in this Response. The calculated values of AIC, BIC, $\chi^2$/dof and $p$-value.

| Energy | Model | AIC | BIC | $\chi^2$/dof | $p$-value |
|---|---|---|---|---|---|
| 1.0–2.5 keV | Extreme solar activity | 347.7317 | 358.3515 | 3.0768 | 0.0 |
| | Very low solar activity | 153.6371 | 164.2570 | 1.2890 | 0.0177 |
| | Severe geomagnetic storm | 704.2318 | 714.8516 | 6.2610 | 0.0 |
| | Quiet geomagnetic activity | 181.7213 | 192.3412 | 1.5636 | $7.3075*10^{-5}$ |
| 2.5–6.0 keV | Extreme solar activity | 50.3180 | 60.9379 | 1.2735 | 0.1177 |
| | Very low solar activity | 53.2962 | 63.9160 | 1.3831 | 0.0562 |
| | Severe geomagnetic storm | 152.6819 | 163.3017 | 3.4441 | $2.1303*10^{-12}$ |
| | Quiet geomagnetic activity | 71.0399 | 81.6597 | 1.9109 | 0.0005 |
| 6.0–10.0 keV | Extreme solar activity | 40.4108 | 51.0307 | 1.1069 | 0.3118 |
| | Very low solar activity | 40.9783 | 51.5981 | 1.1646 | 0.2422 |
| | Severe geomagnetic storm | 42.8781 | 53.4980 | 1.0945 | 0.3282 |
| | Quiet geomagnetic activity | 41.4147 | 52.0346 | 1.1842 | 0.2211 |

The final selection of photon statistics is mainly related to two factors, one is the selection of time bin (binsize) during data reduction (similar to exposure time), the other is the bin of energy channel.

The choice of binsize mainly affects the spatial resolution of the tangent point altitudes ($\delta$ h), that is, the distance between adjacent tangent point altitudes, as shown in Table 3 in this Response. The selection criteria of spatial resolution of tangent point altitudes is that the larger the better, that is, the smaller the distance between adjacent tangent points is, the better, that is, the smaller the binsize is. However, pursuing a smaller binsize can lead to another bad situation, namely, poor photon statistics. Therefore, we need to seek a balance between time bin and photon statistics. The selection criteria is that time bin should not be too large (if it is too large, the spatial resolution of the tangent point altitudes is too small), and photon statistics should not be too small. FIG. 8 in this Response shows the maximum distance, minimum distance and mean distance between adjacent tangent points corresponding to different binsizes. It is found that the distance between adjacent tangent points increases with the increase of binsize.

Table 3 in this Response: The corresponding relationship between time bin (binsize) during data reduction and the spatial resolution of the tangent point altitudes.

| binsize (s) | 0.1 | 0.2 | 0.3 | 0.4 | 0.5 |
|---|---|---|---|---|---|
| Max δh (m) | 207.8282 | 415.5873 | 623.2774 | 836.8984 | 1.0385e+03 |
| Min δh (m) | 123.4646 | 247.0097 | 370.3937 | 494.3417 | 618.1286 |
| Mean δh (m) | 166.6872 | 333.3754 | 499.9465 | 673.7546 | 833.4456 |

| binsize (s) | 0.6 | 0.7 | 0.8 | 0.9 | 1.0 |
|---|---|---|---|---|---|
| Max δh (m) | 1.2459e+03 | 1.4533e+03 | 1.6607e+03 | 1.8680e+03 | 2.0752e+03 |
| Min δh (m) | 741.5126 | 864.5340 | 989.9727 | 1.1123e+03 | 1.2383e+03 |
| Mean δh (m) | 999.9016 | 1.1661e+03 | 1.3335e+03 | 1.4993e+03 | 1.6669e+03 |

[Figure]

FIG. 8 in this Response: Maximum distance, minimum distance, and average distance between adjacent tangent points at different binsizes.

In addition, we calculated the average intensity (mean photon counts) of the unattenuated part of the lightcurves in the energy range of 1.0-2.5 keV, 2.5-6.0 keV and 6.0-10.0 keV corresponding to different binsizes, as shown in Table 4 in this Response. FIG. 9 in this Response shows the average photon counts of unattenuated part of the lightcurves in the energy range of 1.0-2.5 keV, 2.5-6.0 keV and 6.0-10.0 keV for different binsizes.

Table 4 in this Response: The corresponding relationship between time bin (binsize) during data reduction and the unattenuated mean intensity of lightcurves in the energy range of 1.0-2.5 keV, 2.5-6.0 keV and 6.0-10.0 keV.

| binsize (s) | 0.1 | 0.2 | 0.3 | 0.4 | 0.5 |
|---|---|---|---|---|---|
| 1.0-2.5 keV | 40.4351 | 80.8621 | 121.0734 | 161.7500 | 202.1584 |
| 2.5-6.0 keV | 28.1135 | 56.2196 | 84.1766 | 112.4565 | 140.5792 |
| 6.0-10.0 keV | 5.6322 | 11.2686 | 16.8723 | 22.5217 | 28.1357 |

| binsize (s) | 0.6 | 0.7 | 0.8 | 0.9 | 1.0 |
|---|---|---|---|---|---|
| 1.0-2.5 keV | 242.1784 | 281.7799 | 323.4964 | 362.3306 | 404.4595 |
| 2.5-6.0 keV | 168.3946 | 195.9308 | 225.0432 | 252.0565 | 281.2252 |
| 6.0-10.0 keV | 33.7297 | 39.2453 | 45.0360 | 50.4516 | 56.3333 |

[Figure]

FIG. 9 in this Response: Average intensity (mean photon counts) of the unattenuated part of the lightcurves in the energy range of 1.0-2.5 keV, 2.5-6.0 keV and 6.0-10.0 keV under different binsizes.

Through the above discussion, we finally choose binsize=0.4s, because the spatial resolution of the tangent point altitudes corresponding to this value is high enough (the average distance between two adjacent tangent points is about 673 meters), and the lightcurves in the energy range of 1.0-2.5 keV, 2.5-6.0 keV and 6.0-10.0 keV corresponding to this value has relatively high photon statistics. In summary, the signal-to-noise ratio of X-ray photon counting is greater than 4 and the resolution of the tangent point altitude is high enough (less than 1 km),so binsize=0.4 s is finally selected.

In addition, photon counting statistics was also considered in the our fitting. Taking into account the the Poisson nature of our data, the Poisson statistics were used instead of Gaussian statistics in the Markov Chain Monte Carlo sampling. The X-ray counts is different from the detection for traditional ultraviolet, visible and infrared wavelengths. Poisson statistics were used as the likelihood function and C statistics as the logarithmic likelihood function, because C statistics as the logarithmic likelihood function would lead to smaller errors compared with Chi-square statistics[1].

The specific form of Poisson statistics is as follows:

$$\mathcal{L} = \prod_i \frac{M_i^{D_i}}{D_i!} \exp(-M_i) \qquad (2)$$

The specific form of C statistics is as follows:

$$C = 2 \sum_i [M_i - D_i + D_i(\log D_i - \log M_i)] \qquad (3)$$

Reference:

[1] Nousek, J. A. and Shue, D. R., "Chi-squared and C Statistic Minimization for Low Count per Bin Data", The Astrophysical Journal, vol. 342, p. 1207, 1989. doi:10.1086/167676.

---

## Author Comment (AC2)

**In reference to AMT-2021-406 "Measurement of vertical atmospheric density profile from the X-ray Earth occultation of the Crab Nebula with *Insight*-HXMT":**

The author are very grateful to the referees for their valuable comments and suggestions. Our responses to these comments are as follows.

Comments from reviewers:

**Reviewer #1:**

General comments:

This manuscript presents a method to retrieve atmospheric (neutral) density profiles in the mesopause/lower thermosphere region from X-ray occultation observations with Insight-HXMT. The topic of the manuscript is suitable for Atmospheric measurement techniques and novel methods to measure neutral density in the lower thermosphere are certainly of great interest for the scientific community. However, the manuscript contains many linguistical mistakes and little issues. I point out some of them, but probably not all of them. I have several general concerns regarding this paper (which is based on the analysis of a single occultation measurement):

**Reply:** Thanks to the reviewer's comprehensive summary of the work and inspiring comments. The manuscript has been revised carefully based on the received comments. For details, please see the following responses.

1. It took me a while to realize that the method applied here is quite different from the usual methods to retrieve vertical profile information from occultation measurements. If I understand correctly (please correct me if I'm wrong), you don't retrieve the atmospheric density at different altitudes independently, but you simply scale the MSIS density profile by a constant factor. Is this correct?

If yes, I'm not sure what the overall quality of importance of the retrieved density profiles is, because there are potentially large errors associated with this approach. Ideally, you should carry out a vertical profile retrieval as it is done in the many other studies you cited in the introduction, that have not only 1 (or 2) degrees of freedom, but many more. A large part of the paper suggests that you do the "usual" retrieval, which is misleading. If such a simple retrieval (i.e. scaling a model profile) is used, you should at least state explicitly (already in the abstract) that a very basic retrieval is done by simply scaling a model density profile. Ideally, the retrieval should be done as the "standard" occultation retrieval, i.e. by retrieving (more or less) independent information for many different atmospheric layers.

**Reply:** Thank you very much. You are right. Our work is just a simple scaling of the MSIS density profile by the constant correction factor  $\gamma$ . But it should be noted that with this best-fit scale factor we can get the best-fit density profile based on the model density profile as input data. **So the method used in this work provides an approach for the evaluation of the atmospheric models**, e.g, NRLMSISE-00 model and the NRLMSIS 2.0 model, as we did in our work. **We carry out the goodness-of-fit testing for the validation of these measurements.** The null hypothesis can not be rejected even at 84%, 90% confidence level for the two XEO measurements in the energy range of 1.0-2.5keV. The null hypothesis can not be rejected even at 55%, 64% confidence level for the two XEO measurements in the energy range of 6.0-10.0keV. **For further confirmation of these measurements, we also compare the measured density profile with lightcurve fitting to the ones by a standard spectrum retrieval method with an iterative inversion technique as suggested by the reviewer.**

Our retrieval method with lightcurve fitting is an altitude-dependent method. Actually, we also independently retrieve the atmospheric density for different altitude ranges. Because lightcurves of different energies are sensitive to different altitude ranges, atmospheric densities of different altitude ranges can be obtained by fitting the light curves of different energy ranges, but the corresponding altitude of light curves of different energy ranges. For example, the neutral atmospheric densities in the altitude range of 105-200 km can be obtained based on the light curves in the energy range of 1.0-2.5 keV. The neutral atmospheric density in the altitude range of 95-125 km can be obtained based on the light curves in the energy range of 2.5-6.0 keV. The neutral atmospheric densities in the altitude range of 85-110 km can be obtained based on the light curves in the energy range of 6.0-10.0 keV. The measurements with different light curves in different energy range are independent.

We also performed a standard spectrum retrieval with an iterative inversion technique as suggested by the reviewer. Based on the energy spectrum fitting method during X-ray occultation, the altitude independent atmospheric density retrieval results can be obtained, and the overlap of the tangent point altitude can be effectively avoided. In order to prove the reliability of our retrieved results with lightcurve fitting in the paper, we compared our results to the results from energy spectrum fitting. By fitting the energy spectrum data in the energy range of 1-10keV, we obtained the atmospheric density values in the altitude range of 100-200km, and the energy spectra are extracted every 10km. The comparison between the best-fitted model and energy spectrum fitting and the results from lightcurve fitting are shown in FIG. 2 in this Response, where the solid blue line

represents the retrieved results of spectrum fitting, the solid red line represents the model density profile of NRLMSIS 2.0, and the solid green line represents the retrieved results of lightcurve in the energy range of 1.0-2.5 keV. It is found that the fitting results based on the lightcurve are consistent with the retrieved results of the uaual spectrum retrieval method (energy spectrum fitting). The reliability of the measurement results based on lightcurve fitting is validated. Please see section 3.4 in the revised manuscript.

---

## Author Comment (AC3)

**In reference to AMT-2021-406 "Measurement of vertical atmospheric density profile from the X-ray Earth occultation of the Crab Nebula with *Insight*-HXMT":**

The author are very grateful to the referees for their valuable comments and suggestions. Our responses to these comments are as follows.

Comments from reviewers:

**Reviewer #1:**

General comments:

**This manuscript presents a method to retrieve atmospheric (neutral) density profiles in the mesopause/lower thermosphere region from X-ray occultation observations with Insight-HXMT. The topic of the manuscript is suitable for Atmospheric measurement techniques and novel methods to measure neutral density in the lower thermosphere are certainly of great interest for the scientific community. However, the manuscript contains many linguistical mistakes and little issues. I point out some of them, but probably not all of them. I have several general concerns regarding this paper (which is based on the analysis of a single occultation measurement):**

**Reply:** Thanks to the reviewer's comprehensive summary of the work and inspiring comments. The manuscript has been revised carefully based on the received comments. For details, please see the following responses.

**1. It took me a while to realize that the method applied here is quite different from the usual methods to retrieve vertical profile information from occultation measurements. If I understand correctly (please correct me if I'm wrong), you don't retrieve the atmospheric density at different altitudes independently, but you simply scale the MSIS density profile by a constant factor. Is this correct?**

**If yes, I'm not sure what the overall quality of importance of the retrieved density profiles is, because there are potentially large errors associated with this approach. Ideally, you should carry out a vertical profile retrieval as it is done in the many other studies you cited in the introduction, that have not only 1 (or 2) degrees of freedom, but many more. A large part of the paper suggests that you do the "usual" retrieval, which is misleading. If such a simple retrieval (i.e. scaling a model profile) is used, you should at least state explicitly (already in the abstract) that a very basic retrieval is done by simply scaling a model density profile. Ideally, the retrieval should be done as the "standard" occultation retrieval, i.e. by retrieving (more or less) independent information for many different atmospheric layers.**

**Reply:** Thank you very much. You are right. Our work is just a simple scaling of the MSIS density profile by the constant correction factor γ. But it should be noted that with this best-fit scale factor we can get the best-fit density profile based on the model density profile as input data. **So the method used in this work provides an approach for the evaluation of the atmospheric models**, e.g, NRLMSISE-00 model and the NRLMSIS 2.0 model, as we did in our work. **We carry out the goodness-of-fit testing for the validation of these measurements.** The null hypothesis can not be rejected even at 84%, 90% confidence level for the two XEO measurements in the energy range of 1.0-2.5keV. The null hypothesis can not be rejected even at 55%, 64% confidence level for the two XEO measurements in the energy range of 2.5-6.0keV. The null hypothesis can not be rejected even at 68%, 69% confidence level for the two XEO measurements in the energy range of 6.0-10.0keV. **For further confirmation of these measurements, we also compare the measured density profile with lightcurve fitting to the ones by a standard spectrum retrieval method with an iterative inversion technique as suggested by the reviewer.**

Our retrieval method with lightcurve fitting is an altitude-dependent method. Actually, we also independently retrieve the atmospheric density for different altitude ranges. Because lightcurves of different energies are sensitive to different altitude ranges, atmospheric densities of different altitude ranges can be obtained by fitting the light curves of different energy ranges, but the corresponding altitude of light curves of different energy ranges often overlap. For example, the neutral atmospheric densities in the altitude range of 105-200 km can be obtained based on the light curves in the energy range of 1.0-2.5 keV. The neutral atmospheric density in the altitude range of 95-125 km can be obtained based on the light curves in the energy range of 2.5-6.0 keV. The neutral atmospheric densities in the altitude range of 85-110 km can be obtained based on the light curves in the energy range of 6.0-10.0 keV. **The measurements with different light curves in different energy range are independent**.

We also performed a standard spectrum retrieval with an iterative inversion technique as suggested by the reviewer. Based on the energy spectrum fitting method during X-ray occultation, the altitude independent atmospheric density retrieval results can be obtained, and the overlap of the tangent point altitude can be effectively avoided. In order to prove the reliability of our retrieved results with lightcurve fitting in the paper, we compared our results to the results from energy spectrum fitting. By fitting the energy spectrum data in the energy range of 1-10keV, we obtained the atmospheric density values in the altitude range of 100-200km, and the energy spectra are extracted every 10km. The comparison between the best-fitted model and energy spectra data are shown in FIG. 1 in this Response. The retrieved results based on energy spectrum fitting and the results from lightcurve fitting are shown in FIG. 2 in this Response, where the solid blue line

represents the retrieved results of spectrum fitting, the solid red line represents the model density profile of NRLMSIS 2.0, and the solid green line represents the retrieved results of lightcurve in the energy range of 1.0-2.5 keV. It is found that the fitting results based on the lightcurve are consistent with the retrieved results of the uaual spectrum retrieval method (energy spectrum fitting). The reliability of the measurement results based on lightcurve fitting is validated. Please see section 3.4 in the revised manuscript.

[Figure]

FIG. 1 in this Response: Comparison of best-fitted spectrum model and observed spectra data. In the upper space of each panel, blue dots with error bars represent data points, solid red lines represent best-fitted spectrum models, and the lower space of each panel represents residuals between the best fit model and observed spectra data.

[Figure]

FIG. 2 in this Response: Comparison of retrieved results based on energy spectrum fitting and lightcurve fitting. The blue dots with error bars represent the retrieved results with spectrum fitting, the solid red line represents the model density profile of NRLMSIS 2.0, and the solid green line represents the retrieved results with lightcurve fitting in the energy range of 1.0--2.5 keV..

In addition, what is the overall quality of the density profile that you mentioned for our method? We adopted the "standard" process to test the inversion results. We evaluated the measurement uncertainty of the inversion results and tested the hypotheses of the inversion results. Firstly, the goodness of fit of the lightcurve of the best-fitted model and the observed data of lightcurve was tested by chi-square/degree of freedom ($\chi^2/\text{dof}$) and the p-value. As shown in Table 1 in this Response. It is found that the best-fitted model based on retrieved results fits better with the observed data, compared with the MSIS density calculation without scale factor. In addition, it can also be found from the *p*-value that the best fit model based on the retrieved density results fits the observed data better than the model lightcurve based on the MSIS density profiles. Although our retrieved results were only the scaling of the MSIS density profiles, the comparison between the model lightcurve and the observed lightcurve shows that the model lightcurve based on our retrieved results can better fit the observed lightcurve. Thus, the reliability of our inversion results is further confirmed.

Table 1 in this Response: Hypothesis testing results for the extinction curve predictions with XEO measured density profiles and NRLMSISE-00/NRLMSIS 2.0 model simulated density profiles (during the occultation).

| Energy | Method | $\chi^2/\text{dof}$ | *p*-value |
|---|---|---|---|
| 1.0–2.5 keV | XEOS-00 | 1.0599 | 0.1604 |
| | XEOS-2.0 | 1.0756 | 0.1074 |
| | NRLMSISE-00 | 1.1220 | 0.0249 |
| | NRLMSIS 2.0 | 1.0783 | 0.0997 |
| 2.5–6.0 keV | XEOS-00 | 1.0091 | 0.4540 |
| | XEOS-2.0 | 1.0612 | 0.3669 |
| | NRLMSISE-00 | 1.3321 | 0.0802 |
| | NRLMSIS 2.0 | 1.3331 | 0.0797 |
| 6.0–10.0 keV | XEOS-00 | 1.0936 | 0.3293 |
| | XEOS-2.0 | 1.1012 | 0.3193 |
| | NRLMSISE-00 | 1.2085 | 0.1967 |
| | NRLMSIS 2.0 | 1.0955 | 0.3268 |

**2. Please excuse my ignorance, but you write that the X-ray photons are directly absorbed in the K- and L-shells electrons of atoms, including atoms within molecules. Does this mean that, e.g. $O_2$ counts as two absorbing "particles", because both O atoms can absorb? Or does $O_2$ count as one absorbing "particle"? This is not**

**discussed and it will make a big difference. Please discuss this at an appropriate place in the paper.**

**Another general comment: in the introduction you list several existing satellite missions that provide atmospheric "density" profiles. However, most of them product density profiles of specific atmospheric species and not neutral density. Please make sure it is explicitly mentioned what species are retrieved and whether it is (total) neutral density or not.**

**In my opinion the manuscript requires at least a major revision, and ideally a "real" occultation retrieval should be carried out. If I interpreted the method description incorrectly, please let me know.**

**Reply**: Thank you very much. You are right. Although the X-ray photons are directly absorbed in the K- and L-shells electrons of atoms, the $O_2$ counts as one absorbing "particle" in the calculation. The X-ray cross-section data of gas components used in this paper are all from XCOM database. Through calculation, it is found that the X-ray cross section of $O_2$ (or $N_2$) is just two times that of O (or N) atom, so $O_2$ counts as one absorbing "particle" in the calculation, as shown in FIG. 4 in this Response. Please see Line 217-218 in the revised manuscript.

[Figure]

FIG. 4 in this Response: The relationship between the X-ray cross sections of oxygen molecules and oxygen atoms and nitrogen molecules and nitrogen atoms. It is found that the X-ray cross section of $O_2$ (or $N_2$) is just two times that of O (or N) atom. The X-ray cross-section data are obtained from XCOM database.

In the introduction, we mainly mentioned two satellites APOD and TIMED. According to your suggestions, we added the main measurement components of the two satellites. One

of the goals of APOD, a pioneering Project in China, is to measure the total neutral atmospheric density below 520 km. Here, we focus on the SABER instrument on the TIMED satellite, which can observe the vertical distribution of certain atmospheric components, such as ozone, water vapor, carbon dioxide, nitrogen and hydrogen, from the ground to 180 km, but it can also obtain the vertical distribution of total neutral atmospheric density. The total neutral atmospheric density is derived from the ideal gas equation of state combined with the temperature and pressure information obtained by SABER. We had made corresponding modifications in the revised manuscript according to your suggestions.

Specific comments:

**1. Title: I suggest writing "Measurement of the vertical …"**

**Reply:** Thank you very much. "Measurement of vertical ⋯" is modified to "Measurement of the vertical ⋯".

**2. Line 5: Please spell out "HXMT". This is not defined in the entire paper, as far as I can tell.**

**Reply:** Thank you very much. We spelt out "HXMT", which is the Hard X-ray Modulation Telescope. Please see Line 5 on p.1.

**3. Line 18: Please spell out "RXTE"**

**Reply:** We spelt out "RXTE", which is Rossi X-ray Timing Explorer.

**4. Line 20: "study demonstrate" -> "study demonstrates"**

**Reply:** Revised accordingly.

**5. Line 27: "of the reentry vehicle" -> " of reentry vehicles"**

**Reply:** Revised accordingly.

**6. Line 28: "of the reentry vehicle" -> " of reentry vehicles"**

**Reply:** Revised accordingly.

**7. Line 39: "have been being developed" -> "have been developed"**

**Reply:** Revised accordingly.

**8. Line 41: "uesd" -> "used"**

**Reply:** Revised accordingly.

**9. Line 41: "and satellites." Do you really mean in-situ measurements in the middle atmosphere by satellites? I don't think this is possible.**

**Reply:** Thank you very much. It was a mistake. It's hard to measure the middle atmosphere in situ with satellites. but it is possible to do in situ measurements of atmospheric density with satellites at higher altitudes, usually near satellite orbits. Please Line 44 on p.2.

**10. Paragraph starting on line 41: what about falling sphere measurements? They also provide atmospheric density profiles, at least the relative vertical variation of the density.**

**Reply:** We added a description of the falling sphere measurements used to measure the vertical atmospheric density profiles. Please see Line 45-46.

**11. Lines 46 – 49: Please provide more information on the Chinese cubesat project. What altitude range will these in-situ measurements cover? I doubt it is below 130 km or so.**

**Reply:** Thank you very much. Through data review, it was found that the APOD mission was mainly used to measure neutral atmospheric density below 520 km [1]. Please see Line 52.

Reference:

[1] Tang, G., "APOD Mission Status and Observations by VLBI", in New Horizons with VGOS, 2016, pp. 363–367.

**12. Line 53: "and to retrieve atmospheric density"; SABER does retrieve the density profiles of several atmospheric constituents, but I doubt that there is a neutral (total) density data product. Please clarify your statement.**

**Reply:** Thank you very much. SABER can obtain vertical profiles of several atmospheric components, such as $O_3$, $H_2O$, and $CO_2$, as well as neutral atmospheric densities in the altitude range of ~10-140 km. Please see Line 57-58.

**13. Line 55: "In addition to the direct measurements of atmospheric density by sounding rockets and other means"; what do you mean by "other means"? Are there really any other means?**

**Reply:** Thank you very much. "Other means" do create ambiguities. "...other means" is modified to "...satellites and  falling sphere measurements" . Please see Line 60.

**14. Line 58: "There are some previous studies on the retrieval of atmospheric density"; Do you mean specific species or the "total" number density? I think you mean specific species, that should be made clear.**

**Reply:** Thank you very much. That means specific species. "There are some previous studies on the retrieval of atmospheric density" is modified to "There are some previous studies on the retrieval of atmospheric density of specific species". Please see Line 63-64.

**15. Line 61: "basing on" -> "based on"**

**Reply:** Revised accordingly.

**16. Line 65: "used optimal estimation algorithm" -> "used an optimal estimation algorithm"**

**Reply:** Revised accordingly.

**17. Page 3 in general: I'm not sure if you want to provide a complete list of all occultation measurements, but there are several more, i.e. SOFIE/AIM, GOMOS/Envisat, and of course the SAGE and POAM series.**

**Reply:** According to your suggestion, we have added some contents. There are also occultation measurements that invert atmospheric densities for specific species, such as the SOFIE/AIM, GOMOS/Envisat, SAGE series and POAM series. Please see Line 76-79.

**18. Line 81: "However, the Earth's atmospheric density retrieved results are significantly lower" -> "However, the retrieved atmospheric densities are significantly lower"**

**Reply:** Revised accordingly.

**19. Line 83: "temperature profile difference" -> "temperature profile differences"**

**Reply:** Revised accordingly.

**20. Same line: "gravity wave" -> "gravity waves"**

 **Reply:** Revised accordingly.

**21. Line 90: "X-ray" -> "X-rays"**

**Reply:** Revised accordingly.

**22. Line 90: "XEOS can retrieve the neutral atmospheric density in the upper mesosphere and lower thermosphere, which cannot be detected by other means."; I**

don't think this is true. I'm aware of a neutral density retrieval in the MLT region from limb-scatter observations with the SCIAMACHY instrument. This retrieval has not been published, but it is possible to perform these retrievals from limb measurements in the optical spectral range.

**Reply:** Thank you very much. According to your suggestions, we deleted "which cannot be detected by other means".

**23. Line 115: "for the studying of" -> "for the study of"**

**Reply:** Revised accordingly.

**24. General question/comment on the method and Fig. 1: How important is extinction by scattering? Is it negligible compared to absorption? I don't know, and it would be of interest to the reader, I think. Please provide some information on this point.**

**Reply:** X-ray photons are absorbed or scattered by atoms, and the lower the energy, the less significant the scattering effect is relative to the photoelectric absorption effect. In the energy range used in this paper (1-10keV), the scattering effect can be ignored (FIG. 5) because it is too small relative to the photoelectric absorption effect. But in representing extinction, the X-ray cross section we use includes the scattering cross section, as shown in Eq. 2 in the revised manuscript. Please see Line 6-7 in caption in Fig.1 and Line 197-198.

[Figure]

FIG. 5 in this Response: The X-ray cross sections for N, O and Ar. The solid lines represent the total cross section, which refers to the sum of the photoelectric absorption cross section, coherent scattering and incoherent scattering cross section. The dashed

lines represent the photoelectric absorption cross section only. The X-ray cross-section data are obtained from XCOM database.

**25. Caption Fig. 1, line 6: "black solid line" -> "black solid lines"**

**Reply:** Revised accordingly.

**26. Line 130: "Only observations from the small FOV detectors excluding the detector ID of 29 and 87"; Please mention why these detectors were excluded.**

**Reply:** Thank you very much. You are right. We excluded two detectors numbered 29 and 87 because they were damaged. Please see Line 139.

**27. Figure 2: please explain "barn". The typical reader of AMT will probably not know what it means.**

**Reply:** Thank you very much. You are right. A barn is a unit of area equal to $10^{-24}$ cm$^2$. Originally used in nuclear physics for expressing the cross sectional area of nuclei and nuclear reactions. It can be used in all fields of high-energy physics to describe the cross-section. Please Line 3 in caption in Fig.2.

**28. Text around Fig. 3: I suggest mentioning what wavelength range the energy range from 1 - 10 keV corresponds to.**

**Reply:** Thank you very much. You are right. "1 - 10 keV" is modified to "1-10 keV (0.1240 -1.2398nm)". Please see Line 2 in caption in Fig.3.

**29. Figure 4: I'm sorry, but I don't understand this Figure? It doesn't make sense to me. What orbit is the spacecraft in? A LEO, right, according to Fig. 1. The Figure suggests that the Earth is observed from a great distance. Please explain in the caption, how the Figure should be interpreted. How long is t_F?**

**Reply:** Thank you very much. You are right. This is primarily a supplement to explain phenomena such as the ingress and egress. *Insight*-HXMT is a LEO satellite. This figure shows that each orbit has two occultation processes (egress and Ingress). In other words, As a LEO satellite, *Insight* - HXMT can observe the Crab Nebula twice in one orbit. The length of $t_F$ is half the orbital period minus the duration of two occultations, and the orbital period of Insight-HXMT is about 96 minutes. Please see Line 2-3 in caption in Fig.4.

**30. Figure 5: What is the reason for the relatively large variability above 100 km tangent height?**

**Reply:** Thank you very much. The reason for the relatively large variability of the light curve above 100km is the absorption of X-ray photons by atoms of atmospheric

components. X-ray photons in this energy range are sensitive to this altitude range and can then be retrieved for total neutral atmospheric density in this range. Please see Line 7-8 in Fig.5.

**31. Caption Fig. 5, line 1: "are observations data" -> "are observation (or observational) data"**

**Reply:** Revised accordingly.

**32. Same caption, line 2: "represent the trend"; How was the "trend" determined? "Trend" is a very vague term here.**

**Reply:** Thank you very much. The concept of "trend" is indeed vague, and we changed this "trend" into "modelled light curve". Please see Line 2 in caption in Fig.5.

**33. Same line: "regions correspond" -> "region corresponds"**

**Reply:** Revised accordingly.

**34. Same caption, line 3 AND line 5: "height ranges" -> "height range"**

**Reply:** Revised accordingly.

**35. Same caption, line 4: "The blue shadow colored regions correspond" -> "The blue colored region corresponds"**

**Reply:** Revised accordingly.

**36. Same line: "For clarity, the extinction process for occultation"; please rephrase, it is not the "extinction process" that is shown here, but the height range, where it is relevant.**

**Reply:** Thank you very much. "extinction process" is modified to "height range".

**37. Line 155: "represent the trend"; What does "trend" mean here? What function is used?**

**Reply:** Thank you very much. You are right. The concept of "trend" is indeed vague, and we changed this "trend" into "modelled light curve". Please see Line 163.

**38. Line 157: "height ranges" -> "height range"**

**Reply:** Revised accordingly.

**39. Line 163: Please delete "X-ray" in "for X-ray atmospheric density".**

**Reply:** Revised accordingly.

**40. Line 164: "The ionized states, electronic states and chemical bonds within the molecules of atmospheric components have no effect on the absorption of X-rays in the extinction process."; What about O2 (or N2). Does O2 count as 2 absorbing "particles" or as one? If each atom counts, then one has to make assumptions on the relative abundance of atomic and molecular constituents (O vs. O2 and N vs. N2) in the retrieval.**

**Reply:** Thank you very much. You are right. Although the X-ray photons are directly absorbed in the K- and L-shells electrons of atoms, the $O_2$ counts as one absorbing "particle" in the calculation. The X-ray cross-section data of gas components used in this paper are all from XCOM database. Through calculation, it is found that the X-ray cross section of $O_2$ (or $N_2$) is just two times that of O (or N) atom, so $O_2$ counts as one absorbing "particle" in the calculation, as shown in FIG. 6 in this Response. Please see Line 217-218 in the revised manuscript.

[Figure]

FIG. 6 in this Response: The relationship between the X-ray cross sections of oxygen molecules and oxygen atoms and nitrogen molecules and nitrogen atoms. It is found that the X-ray cross section of $O_2$ (or $N_2$) is just two times that of O (or N) atom. The X-ray cross-section data are obtained from XCOM database.

**41. Line 168: "It is impossible to distinguish atoms from molecules"; please see last comment.**

**Reply:** Thank you very much. You are right. Although the X-ray photons are directly absorbed in the K- and L-shells electrons of atoms, the $O_2$ counts as one absorbing "particle" in the calculation. The X-ray cross-section data of gas components used in this paper are all from XCOM database. Through calculation, it is found that the X-ray cross

section of $O_2$ (or $N_2$) is just two times that of O (or N) atom, so $O_2$ counts as one absorbing "particle" in the calculation, as shown in FIG. 7 in this Response. Please see Line 217-218 in the revised manuscript.

[Figure]

FIG. 7 in this Response: The relationship between the X-ray cross sections of oxygen molecules and oxygen atoms and nitrogen molecules and nitrogen atoms. It is found that the X-ray cross section of $O_2$ (or $N_2$) is just two times that of O (or N) atom. The X-ray cross-section data are obtained from XCOM database.

**Caption Fig. 7, line 1: "The predicted lightcurves from Insight-HXMT"; Are they predicted (i.e. modelled) or "from" the measurements? Please clarify.**

**Reply:** Thank you very much. "The predicted lightcurves from *Insight*-HXMT" is modified to "The modelled lightcurves from *Insight*-HXMT".

**42. Caption Fig. 7, line 3: "basis function"; I'm not sure why the term "function" is used here? I'd use "input data" or something like that.**

**Reply:** Thank you very much. The term "basis function" is really a misnomer, but here we want to express the input density profile from the MSIS model. According to your suggestions, "basis function" is modified to "input data".

**43. Caption Fig. 7, line 4: "occultation depth"; What do you mean by "occultation depths". Please rephrase or define.**

**Reply:** "occultation depth" represents the difference between the highest and lowest point of the same light curve. Please see Line 4 in Fig.7.

**44. Equation (2): I don't really understand this equation. The integral is the integral along the line of sight, right? If yes, this should be mentioned explicitly. If you integrate along the line of sight, why is N_s the column density of each component along the line of sight. That doesn't make sense. I think something is wrong here.**

**Reply:** Thank you very much. You are right. $N_s$ should be the number density of each component along the line of sight. At the same time, we changed $N_s$ to $n_s$ to represent the number density. Please see Line 193 in the revised manuscript.

**I'm also not sure, why the correction factor gamma_s is needed here. The optical depth is determined by number densities (or the integral thereof) of the relevant species and the cross section. At this stage, not correction should be required.**

**Reply:** Thank you very much. Here, $\gamma_s$ represents the correction factor, which is a key step in building the forward model for the lightcurve modeling. The correction factor $\gamma_s$ reflects the difference between the real number density and the input data of the number density with NRLMSISE-00/NRLMSIS 2.0 model calculations in our work. Here, we want to obtain the difference between the real number density and the number density with NRLMSISE-00/NRLMSIS 2.0 model calculations, i.e., obtain the best-fitted number density. Therefore, the correction factor $\gamma_s$ cannot be omitted.

**I think that the number densities here are the MSIS model number densities, which are corrected by the correction factor. Is this the case? This must be mentioned explicitly, otherwise one cannot understand what is done here.**

**Reply:** Thank you very much. You are right. Yes, we only scale the density profile of the MSIS model density through the correction factor, and we further clarify this fact. Please see Line 193 in the revised manuscript.

**Also: There is no mention on the ray-tracing through the spherical atmosphere, i.e. the determination of the slant column densities. How do you do this? What assumptions is it based on?**

**Reply:** Thank you very much. You are right. Based on the spherical symmetry assumption of the Earth atmosphere, the number density is converted to column density by Abelian integral. Please see Line 195-196 in the revised manuscript.

**45. Line 187: "is shown" -> "are shown"**

**Reply:** Revised accordingly.

**46. Line 187: ".. from Insight-HXMT"; Are they predicted (i.e. modelled) or "from" the measurements? Please clarify.**

**Reply:** Thank you very much. "predicted " is modified to "modelled". Please see Line 200 in the revised manuscript.

**47. Line 188: Please define "occultation depths" or use another term.**

**Reply:** Thank you very much. You are right. Please see Line 202 in the revised manuscript.

**48. Line 190: "our basis function"; again, why "function"? Is it really a function in the mathematical sense?**

**Reply:** Thank you very much. The term "basis function" is really a misnomer, but here we want to express the input density profile from the MSIS model. According to your suggestions, "basis function" is modified to "input data". Please see Line 203 in the revised manuscript.

**49. Line 195: "fitting is good" -> "fit is good"**

**Reply:** Revised accordingly.

**50. Caption Fig. 8, line 2: "of the fitting" -> "of the fit"**

**Reply:** Revised accordingly.

**51. Line 198: "of the fitting" -> "of the fit"**

**Reply:** Revised accordingly.

**52. Line 202: "but their total atmospheric density (N+O)"; Does "N" here stand for N and N2? This should be made clear.**

**Reply:** You are right. "N" here stand for N and $N_2$. "(N+O)" is modified to "(N+O+$O_2$+$N_2$)". Please see Line 216 in the revised manuscript.

**53. Line 203: "Ar is an atmospheric composition" -> "Ar is an atmospheric constituent"**

**Reply:** Revised accordingly.

**54. Line 207: "The number density of each atmospheric component needs to be given as a basis function"; What does basis function mean here? Does function mean a parametrization of the vertical density variation? This is not clear.**

**Reply:** Thank you very much. The term "basis function" is really a misnomer, but here we want to express the input density profile from the MSIS model. According to your

suggestions, "basis function" is modified to "input data". Please see Line 203 in the revised manuscript. Please see Line 223 in the revised manuscript.

**55. Line 208: "the atmospheric model" -> "the atmospheric models"**

**Reply:** Revised accordingly.

**56. Line 209: "as our basis function"; see above comment.**

**Reply:** Thank you very much. The term "basis function" is really a misnomer, but here we want to express the input density profile from the MSIS model. According to your suggestions, "basis function" is modified to "input data". Please see Line 203 in the revised manuscript. Please see Line 225 in the revised manuscript.

**57. Table 3, column 4: what does "average" mean here for the F10.7 cm radio flux? Over what time or spatial range did you average and why?**

**Reply:** Thank you very much. You are right. "average" mean that 81 day average of F10.7 flux, centered on day of year. We calculated the MSIS density as input data by using the MATLAB function *atmosnrlmsise00*. The average of F10.7 cm radio flux is one of the inputs of the MATLAB function *atmosnrlmsise00*. See the links below for detailed descriptions: Implement mathematical representation of 2001 United States Naval Research Laboratory Mass Spectrometer and Incoherent Scatter Radar Exosphere - MATLAB atmosnrlmsise00 (mathworks.com)

**58. Equation (3): Shouldn't the background noise B be zero on average? Why is this not the case?**

**Reply:** Thank you very much. In the process of fitting, we carry out fitting with the background. This background contains the diffuse X-ray radiation from the universe. Although the background of LE is very small, it is still not zero.

**59. Equation (5): "!" at the end; Is this intentional, i.e. is this a factorial? of D_i?**

**Reply:** Thank you very much. This is the factorial[1].

Reference:

[1]Cash, W., "Parameter estimation in astronomy through application of the likelihood ratio.", The Astrophysical Journal, vol. 228, pp. 939–947, 1979. doi:10.1086/156922.

**60. Line 247: "for the correction factor gamma_s"; Please state explicitly, whether the entire MSIS density profile is scaled with this single factor, or whether the factor depends on altitude.**

**Reply:** Thank you very much. You are right. The retrieved results are a simple scaling of the MSIS density profile by a correction factor. Please see Line 271-273 in the revised manuscript.

**61. Line 250: "where the first 1000 steps in each walker are burned."; What does "burned" mean here?**

**Reply:** Thank you very much. Here "burned" means that throw away. We throw away or delete the non-convergent Markov Chains. As shown in FIG. 7 in this Response. This is a Markov Chain of 2000 sample points in FIG 2 in this Response, which obviously does not converge at the first about 100 sample points and therefore must be throw away, and of course it doesn't matter if you throw away more (e.g. the first 1,000 steps), because the rest of the Markov Chain has converged.

[Figure]

FIG. 7 in this Response: The first 2,000 steps of the Markov Chain for the two model parameter.

**62. Line 256: "basis functions"; see above comments.**

**Reply:** Revised accordingly.

**63. Line 276: "This indicates an overestimation for the density from NRLMSISE-00 model prediction."; Only if the NRLMSIS 2.0 result is correct, which we don't know. I suggest deleting this statement. The difference between the model versions is already stated in the previous sentence.**

**Reply:** Thank you very much. You are right. "This indicates an overestimation for the density from NRLMSISE-00 model prediction." is deleted.

**64. Line 305: Please mention, how many degrees of freedom (dof) are there. Also: "dof" has not been defined.**

**Reply:** The degree of freedom is defined as follows: dof = n-k, where dof represents the degree of freedom, n represents the number of sample points used for fitting, and k represents the number of variables. Here, k=2, because there are two variables: correction factor and background noise, n=551, because there are 551 sample points used for fitting, so dof =549. Please see Line 321-323 in the revised manuscript.

**65. Line 320: "and the gaps" -> "and gaps"**

**Reply:** Revised accordingly.

**66. Line 333: "altitude range .. overlaps" -> „altitude ranges .. overlap"**

**Reply:** Revised accordingly.

**67. Line 337: "This is because the XEOS method is an altitude-dependent method, different energy bands have different sensitive altitude ranges .."; It may also be a consequence of your basic approach to use a scaling factor (if I understand correctly) rather than retrieving the actual vertical variation.**

**Reply:** Thank you very much. You are right. The XEOS method represent the method of lightcurve fitting we used in this paper. The retrieved results with lightcurve fitting are a simple scaling of MSIS density. But we can retrieve the actual vertical variation with the standard energy spectrum fitting. Please see Line 355-356 and section 3.4 in the revised manuscript.

**68. Line 341: "by solar activities and geomagnetic activities" -> "by solar activity and geomagnetic activity"**

**Reply:** Revised accordingly.

**69. Line 343: Same comment**

**Reply:** Revised accordingly.

**70. Line 346: delete "the" in "under the extreme" and "and the very"**

 **Reply:** Revised accordingly.

**71. Same line: "the severe geomagnetic storm" -> "a severe geomagnetic storm"**

**Reply:** Revised accordingly.

**72. Line 347: delete "the" in "and the quiet .."**

**Reply:** Revised accordingly.

**73. Line 351: Please explain AIC and BIC. What do the numbers mean? These criteria are not discussed in the paper so far.**

**Reply:** AIC and BIC are used for model selection, and can also be used to compare models. Usually, we choose the model with the smallest AIC and BIC. However, the values of AIC and BIC in this paper show that solar and geomagnetic activity has a great influence on model shape. Because AIC and BIC values vary greatly under different solar and geomagnetic activity. Please see Line 370-373 in the revised manuscript.

**74. Line 361: "the density profile is retrieved."; Please state explicitly, whether only a scaled version of the MSIS density profile is "retrieved" or a vertical density profile with more than one degree of freedom.**

**Reply:** You are right. The retrieved results with lightcurve fitting are only a simple scaling of the MSIS density profile. A vertical density profile with more than one degree of freedom can also be retrieved with the standard energy spectrum fitting method. We compared the lightcurve fitted results to the energy spectrum fitted ones, and they are consistent with each other. Please see Line 384 and section 3.4 in the revised manuscript.

**75. Line 366: "And the extinction curve can be better described by the XEO retrieved density profile."; I don't think this has to be mentioned explicitly. That should be obvious. If it is not the case there is something wrong with the retrieval.**

**Reply:** Thank you very much. "And the extinction curve can be better described by the XEO retrieved density profile." is deleted.

**76. Line 371: "The XEO retrieved density profile in the altitude range of 95–125 km has a better description for the XEO extinction lightcurve than the NRLMSISE-00/NRLMSIS 2.0 model .."; Again, this does not have to be mentioned explicitly. Please delete.**

**Reply:** Thank you very much. "The XEO retrieved density profile in the altitude range of 95-125 km has a better description for the XEO extinction lightcurve than the NRLMSISE-00/NRLMSIS 2.0 model predicted density profile at the same date, time and geographical location." is deleted.

**77. Line 383: "The Insight-HXMT satellite can join the family of the XEOS."; Well, you should first do a full vertical profile retrieval, not just a scaling.**

**Reply:** Thank you very much. In order to avoid the ambiguity of the method compared with the traditional occultation method, we deleted "The Insight-HXMT satellite can join the family of the XEOS.".

**78. First text block on page 18: One can use the information from different spectral channels in a simultaneous retrieval. This should be the goal.**

**Reply:** Thank you very much. You are right. We added "The goal is that One can use the information from different spectral channels in a simultaneous retrieval.". Please see Line 405 in the revised manuscript.

**79. Line 393: The differences can also be due to model errors and/or retrieval errors.**

**Reply:** Thank you very much. You are right. We added "And the differences can also be due to model errors and/or retrieval errors.". Please see Line 413 in the revised manuscript.

**80. Line 395: "gravitational waves"; I doubt it. You mean gravity waves, not gravitational waves, right.**

**Reply:** Revised accordingly. "gravitational waves" is modified to "gravity waves".

**81. Lines 391 and 395: add space before Determan reference.**

**Reply:** Revised accordingly.

**82. Line 398: "retrieval method need" -> „retrieval methods needs"**

**Reply:** Revised accordingly.

**83. Figure 11: Why do you ratio the HXMT results and the MSIS profiles for Sep 30, 2018 by the MSIS profile on Nov 14 2005? That doesn't make sense in my opinion. It would make more sense to use an MSIS profile for Sep 30, 2018, because this date is the focus of the current paper.**

**Reply:** Thank you very much. You are right. The MSIS profile on Sep 30, 2018 is indeed more meaningful than the MSIS profile on Nov 14, 2005. However, the inversion results based on RXTE came from November 14, 2005, so we drew the MSIS profile on November 14, 2005 in the figure, and normalized based on it. So both MSIS profiles on Nov 14, 2005 and Sep 30, 2018 are included.

**84. Figure 13: Please mention in Figure caption, which model was used for these simulations.**

**Reply:** Thank you very much. You are right. We added "The NRLMSISE-00 density profiles are used as input data for those simulations." in the caption in Figure 13.